# Inhibition of DNMT1 methyltransferase activity via glucose-regulated *O*-GlcNAcylation alters the epigenome

**Heon Shin[1], Amy Leung[1], Kevin R Costello[1,2], Parijat Senapati[1], Hiroyuki Kato[1], Roger E Moore[3], Michael Lee[1,2], Dimitri Lin[1], Xiaofang Tang[1], Patrick Pirrotte[1,3,4], Zhen Bouman Chen[1,2], Dustin E Schones[1,2]\***

[1]Department of Diabetes Complications and Metabolism, Beckman Research Institute, City of Hope, Duarte, United States; [2]Irell and Manella Graduate School of Biological Sciences, City of Hope, Duarte, United States; [3]Integrated Mass Spectrometry Shared Resource, City of Hope Comprehensive Cancer Center Duarte, Duarte, United States; [4]Cancer & Cell Biology Division, Translational Genomics Research Institute, Phoenix, United States

**Abstract** The DNA methyltransferase activity of DNMT1 is vital for genomic maintenance of DNA methylation. We report here that DNMT1 function is regulated by *O*-GlcNAcylation, a protein modification that is sensitive to glucose levels, and that elevated *O*-GlcNAcylation of DNMT1 from high glucose environment leads to alterations to the epigenome. Using mass spectrometry and complementary alanine mutation experiments, we identified S878 as the major residue that is *O*-GlcNAcylated on human DNMT1. Functional studies in human and mouse cells further revealed that *O*-GlcNAcylation of DNMT1-S878 results in an inhibition of methyltransferase activity, resulting in a general loss of DNA methylation that preferentially occurs at partially methylated domains (PMDs). This loss of methylation corresponds with an increase in DNA damage and apoptosis. These results establish *O*-GlcNAcylation of DNMT1 as a mechanism through which the epigenome is regulated by glucose metabolism and implicates a role for glycosylation of DNMT1 in metabolic diseases characterized by hyperglycemia.

\*For correspondence: dschones@coh.org

**Competing interest:** The authors declare that no competing interests exist.

## Editor's evaluation

In this study Shin and colleagues investigate the role of the posttranslational modification of the DNA methyltransferase by covalent linkage of the N-Acetylglucosamine (O-GlcNAc). The authors present compelling evidence showing that a prolonged high fat/sucrose diet causes global protein O-GlcNAcylation in the liver and DNMT1 is among the proteins that increase their O-GlcNAc level. This result is significant because of the paucity of in vivo data addressing the interplay between metabolism and protein O-GlcNAcylation. The paper also shows that DNMT1's O-GlcNAcylation level correlated to extracellular glucose levels in other cell types.

## Introduction

Protein *O*-GlcNAcylation is a dynamic and reversible post-translational modification that attaches a single *O*-linked β-*N*-acetylglucosamine to serine or threonine residues (*Hart et al., 1996*). It is modulated by two *O*-GlcNAc cycling enzymes, *O*-GlcNAc transferase (OGT) and *O*-GlcNAcase (OGA), that respond to metabolic signals (*Hart et al., 2007*; *Slawson et al., 2010*). Increased concentrations of UDP-GlcNAc that are observed in conditions of excess glucose lead to a general increase in protein

*O*-GlcNAcylation (*Walgren et al., 2003*). Obesogenic diets, furthermore, have elevated protein *O*-GlcNAcylation in various human cell types, including liver cells (*Guinez et al., 2011*), lymphocytes (*Torres and Hart, 1984*), and immune cells (*de Jesus et al., 2018*).

As with other post-translational modifications, *O*-GlcNAcylation of proteins can influence the function and/or stability of the targeted proteins (*Shin et al., 2018*; *Yang and Qian, 2017b*). Thousands of proteins are targets for *O*-GlcNAcylation, including many epigenetic regulatory proteins. For example, the *O*-GlcNAcylation of TET family proteins alters their activity, localization, and targeting (*Chen et al., 2013*; *Ito et al., 2014*; *Shi et al., 2013*; *Zhang et al., 2014*). While all DNA methyltransferases have been shown to be *O*-GlcNAcylated (*Boulard et al., 2020*), the functional consequences of this have not been previously investigated.

Among the DNA methyltransferase (DNMT) family of proteins, DNMT1 is imperative for maintaining DNA methylation patterns during replication (*Bestor and Ingram, 1983*). DNMT1 is a modular protein with several domains necessary for interacting with cofactors, including the BAH1 and BAH2 domains (*Maresca et al., 2015*; *Ren et al., 2018*). The stability and function of DNMT1 have been shown to be regulated through post-translational modifications, including acetylation, phosphorylation, and methylation (*Scott et al., 2014*).

Partially methylated domains (PMDs), large domains with a loss of DNA methylation, were originally identified in cultured cell lines (*Lister et al., 2009*) and subsequently found to be a characteristic of cancer cells (*Berman et al., 2011*; *Brinkman et al., 2019*). PMDs have also been detected in noncancerous healthy tissues, where they are associated with late replication loci (*Hansen et al., 2010*; *Zhou et al., 2018*). While PMDs are generally thought to arise from a lack of fidelity in maintenance methylation (*Decato et al., 2020*), the mechanisms responsible for the establishment of PMDs have remained unclear. Here, we report that the activity of DNMT1 is regulated by extracellular levels of glucose through *O*-GlcNAcylation and is associated with loss of methylation within PMDs.

## Results
### High glucose conditions increase *O*-GlcNAcylation of DNMT1
To validate that DNMT1 can be *O*-GlcNAcylated, we treated Hep3B cells with OSMI-4 (OSMI), an OGT inhibitor (*Martin et al., 2018*), or with Thiamet-G (TMG), an OGA inhibitor (*Elbatrawy et al., 2020*). As expected, immunoblots of cellular lysate with an antibody recognizing pan-*O*-GlcNAc (RL2) reveal that inhibition of OGA increased global levels of *O*-GlcNAc while inhibition of OGT decreased global levels of *O*-GlcNAc (*Figure 1—figure supplement 1*). To distinguish whether DNMT1 is *O*-Glc-NAcylated, DNMT1 immunoprecipitation was performed with cellular lysates treated with OSMI or TMG. Immunoblots with *O*-GlcNAc antibodies revealed that TMG treatment increases *O*-GlcNAc of DNMT1 while OSMI treatment decreases *O*-GlcNAc (*Figure 1—figure supplement 1*). In addition to Hep3B cells, we found that DNMT1 is *O*-GlcNAcylated in HepG2 cells (*Figure 1—figure supplement 2*) and B-cell-derived lymphocytes, indicating that DNMT1 is *O*-GlcNAcylated across various cell types (*Figure 1—figure supplement 2*).

To assess the effect of increased glucose metabolism on *O*-GlcNAcylation of DNMT1, we treated Hep3B cells with normal or low concentrations of glucose (5 mM) or high glucose (25 mM) and examined global protein *O*-GlcNAcylation as well as the *O*-GlcNAcylation of DNMT1 specifically (*Hardivillé et al., 2020*). Consistent with previous reports (*Andrews et al., 2000*), the total amount of protein *O*-GlcNAcylation was increased with high glucose treatment (*Figure 1A*). Global protein *O*-GlcNAcylation was also induced with high concentrations of sucrose, albeit to a lower extent than with glucose (*Figure 1—figure supplement 3*). To specifically assess the level of *O*-GlcNAcylated DNMT1, we performed immunoprecipitation of DNMT1 from lysates of glucose treated Hep3B cells and immunoblotted for *O*-GlcNAc. As with the analysis of total protein, high glucose treatment increased the *O*-GlcNAcylation of DNMT1 (*Figure 1B*). High sucrose treatment also increased the *O*-GlcNAcylation of DNMT1 (*Figure 1—figure supplement 3*). The increased *O*-GlcNAcylation of DNMT1 with high glucose and sucrose treatment was also observed in HepG2 cells (*Figure 1—figure supplement 3*). The changes in *O*-GlcNAcylation of DNMT1 from glucose treatment were not influenced by alterations in the activity of OGT or OGA as we observed the enzymatic activity of OGT or OGA was not significantly changed by glucose treatment (*Figure 1—figure supplement 4*), consistent with previous results (*Seo et al., 2016*).

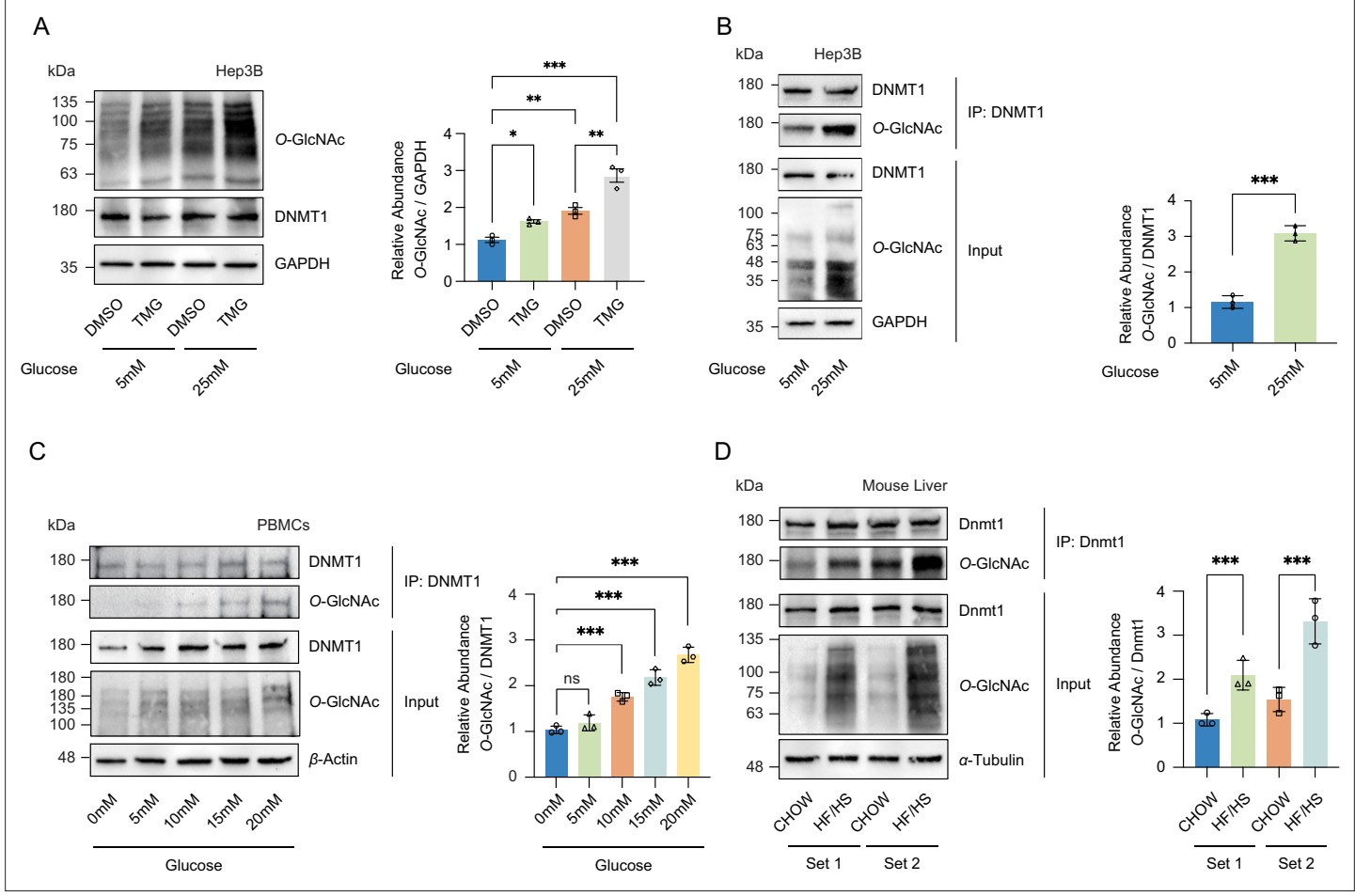

**Figure 1.** High glucose increases *O*-GlcNAcylation of DNMT1 in cell lines and primary cells. (**A**) Hep3B cells were treated with glucose (5 mM or 25 mM) with or without Thiamet-G (TMG). Shown are immunoblots of collected lysates using antibody targeting *O*-GlcNAc and GAPDH (n = 3). (**B**) Lysates of Hep3B treated with glucose were immunoprecipitated with DNMT1 and immunoprecipitates were immunoblotted with antibody targeting *O*-GlcNAc (n = 3). (**C**) Peripheral blood mononuclear cells (PBMCs) were isolated from three individual donor blood samples and treated with increasing concentration of glucose for 24 hr. Collected cell lysates from PBMCs were immunoprecipitated with antibody targeting DNMT1 and immunoblotted for *O*-GlcNAc. Representative blot from one donor (n = 3). (**D**) Immunoblots for *O*-GlcNAc and GAPDH from liver samples of C57BL/6J mice given a high-fat/high-sucrose diet (HF/HS) or normal diet (chow) for 4 mo, and immunoprecipitated with Dnmt1. Lysates of mouse liver were immunoprecipitated with Dnmt1 and immunoprecipitates were immunoblotted with antibody targeting *O*-GlcNAc. *p<0.001; **p<0.0005; ***p<0.0001 by Student's *t*-test (**A-D**); ns, not significant; data are represented as mean ± SD from three replicates of each sample.

The online version of this article includes the following source data and figure supplement(s) for figure 1:

**Source data 1.** Uncropped blot files of *Figure 1A–D*.

**Figure supplement 1.** DNMT1 can be *O*-GlcNAcylated in Hep3B cells.

**Figure supplement 1—source data 1.** Uncropped blot files of *Figure 1—figure supplement 1*.

**Figure supplement 2.** DNMT1 can be *O*-GlcNAcylated in HepG2 cells and B-cells-derived lymphocytes.

**Figure supplement 2—source data 1.** Uncropped blot files of *Figure 1—figure supplement 2A and B*.

**Figure supplement 3.** Global protein *O*-GlcNAcylation was induced with high concentrations of sucrose.

**Figure supplement 3—source data 1.** Uncropped blot files of *Figure 1—figure supplement 3A and B*.

**Figure supplement 4.** The enzymatic activity of OGT or OGA was not significantly changed by glucose treatment.

**Figure supplement 5.** DNMT1 can be *O*-GlcNAcylated in primary cells (peripheral blood mononuclear cells [PBMCs]).

**Figure supplement 5—source data 1.** Uncropped blot files of *Figure 1—figure supplement 5*.

To examine whether an increase in *O*-GlcNAcylation of DNMT1 also occurs in primary cells, we collected peripheral blood mononuclear cells (PBMCs) from three separate patient donors and treated the PBMCs with increasing glucose levels (0 mM, 5 mM, 10 mM, 15 mM, and 20 mM). Consistent with our observations in Hep3B cells, we observed an increase in *O*-GlcNAcylation of DNMT1 with increased glucose levels (*Figure 1C*). Combining the high glucose condition with OGA inhibition by TMG resulted in a further increase in *O*-GlcNAcylation of DNMT1 (*Figure 1A* and *Figure 1—figure supplement 5*). To examine the relationship between glucose levels and the *O*-GlcNAcylation of DNMT1 in an in vivo context, we examined liver samples from C57BL/6J mice fed an obesogenic high-fat/high-sucrose (HF/HS) diet for 16 wk (*Tang et al., 2020*; details in 'Materials and methods'). These samples displayed an increase in total *O*-GlcNAcylation in liver samples from HF/HS fed mice (*Figure 1D*) as well as increased *O*-GlcNAcylation of DNMT1 (*Figure 1D*). These data validate the *O*-GlcNAcylation of DNMT1 and that the degree of *O*-GlcNAcylation of DNMT1 increases with glucose concentrations.

## Identification of the major *O*-GlcNAcylation sites of DNMT1

To begin to identify the residue(s) *O*-GlcNAcylated on DNMT1, we utilized OGTSite (*Kao et al., 2015*) to predict potential sites of *O*-GlcNAcylation. OGTSite, which uses experimentally verified *O*-GlcNAcylation sites to build models of substrate motifs, identified 16 candidate *O*-GlcNAc modified sites on human DNMT1 (*Supplementary file 1*). We next employed mass spectrometry analysis to examine the post-translational modifications on DMNT1 in Hep3B cells. We overexpressed DNMT1 using a Myc-tagged DNMT1 construct to increase the protein level of DNMT1 in Hep3B cells (*Li et al., 2006*). Immunoblots with Myc antibody (*Yompakdee et al., 1996*) revealed a band corresponding to Myc-DNMT1 in transfected, but not mock transfected, cells (*Figure 2—figure supplement 1*). We further confirmed with immunoprecipitation followed by immunoblot that the overexpressed Myc-DNMT1 can be *O*-GlcNAcylated (*Figure 2—figure supplement 1*). For mass spectrometry analysis, we treated Myc-DNMT1 expressing cells with 25 mM TMG to further increase the *O*-GlcNAcylation of DNMT1. Myc-DNMT1 was enriched from transfected cells by monoclonal Ab-crosslinked immunoprecipitation and subjected to in-solution digestion using three different enzymes (AspN, chymotrypsin, and LysC) and high-resolution LC-MS/MS analysis. Peptide analyses revealed that S878, which is located on the bromo-associated homology (BAH1) domain of DNMT1 is *O*-GlcNAcylated (*Figure 2A and B*, *Figure 2—figure supplement 2*, and *Supplementary file 2*). Analysis of the evolutionary conservation of DNMT1 sequence (details in 'Materials and methods') revealed that S878 is conserved through vertebrates to *Arabidopsis thaliana* (*Figure 2C*). Three previously unreported phosphorylated residues were also detected (T208, S209, and S1122) (*Supplementary files 2 and 3*).

We chose the three top candidates based on prediction score (T158, T616, and T882) as well as the site identified from mass spectrometry analysis (S878) for further analysis with alanine mutation experiments (*Rexach et al., 2012*). The threonine/serine residues were mutated to alanine residues on the Myc-DNMT1 construct and *O*-GlcNAcylation was evaluated with immunoblot following immunoprecipitation. Loss of threonine and serine at positions T158 and S878 respectively resulted in a loss of *O*-GlcNAcylation, indicating that these two residues are required for *O*-GlcNAcylation, with the DNMT1-S878A and DNMT1-T158A/S878A mutant resulting in >50% reduction of *O*-GlcNAcylation (*Figure 2D*). These results indicate that T158 (near the PCNA binding domain) and S878 (within the BAH1 domain) are the *O*-GlcNAcylated residues of DNMT1.

## *O*-GlcNAcylation of DNMT1 results in loss of DNA methyltransferase activity

The BAH domains of DNMT1 are known to be necessary for DNA methyltransferase activity (*Gong et al., 2021*; *Yarychkivska et al., 2018*). Given that S878 is in the BAH1 domain, we reasoned that *O*-GlcNAcylation of this residue could impact the DNA methyltransferase activity of DNMT1. To test this, we treated Hep3B and HepG2 cells with either low (5 mM, CTRL) or high glucose combined with TMG (25 mM, *O*-GlcNAc) and evaluated the DNA methyltransferase activity of immunoprecipitated DNMT1 with the EpiQuik DNMT Activity/Inhibition ELISA Easy Kit (P-3139, EpiGentek, details in 'Materials and methods'). Intriguingly, high glucose/TMG treatment reduced the activity of DNMT1 (*Figure 3A* and *Figure 3—figure supplement 1*).

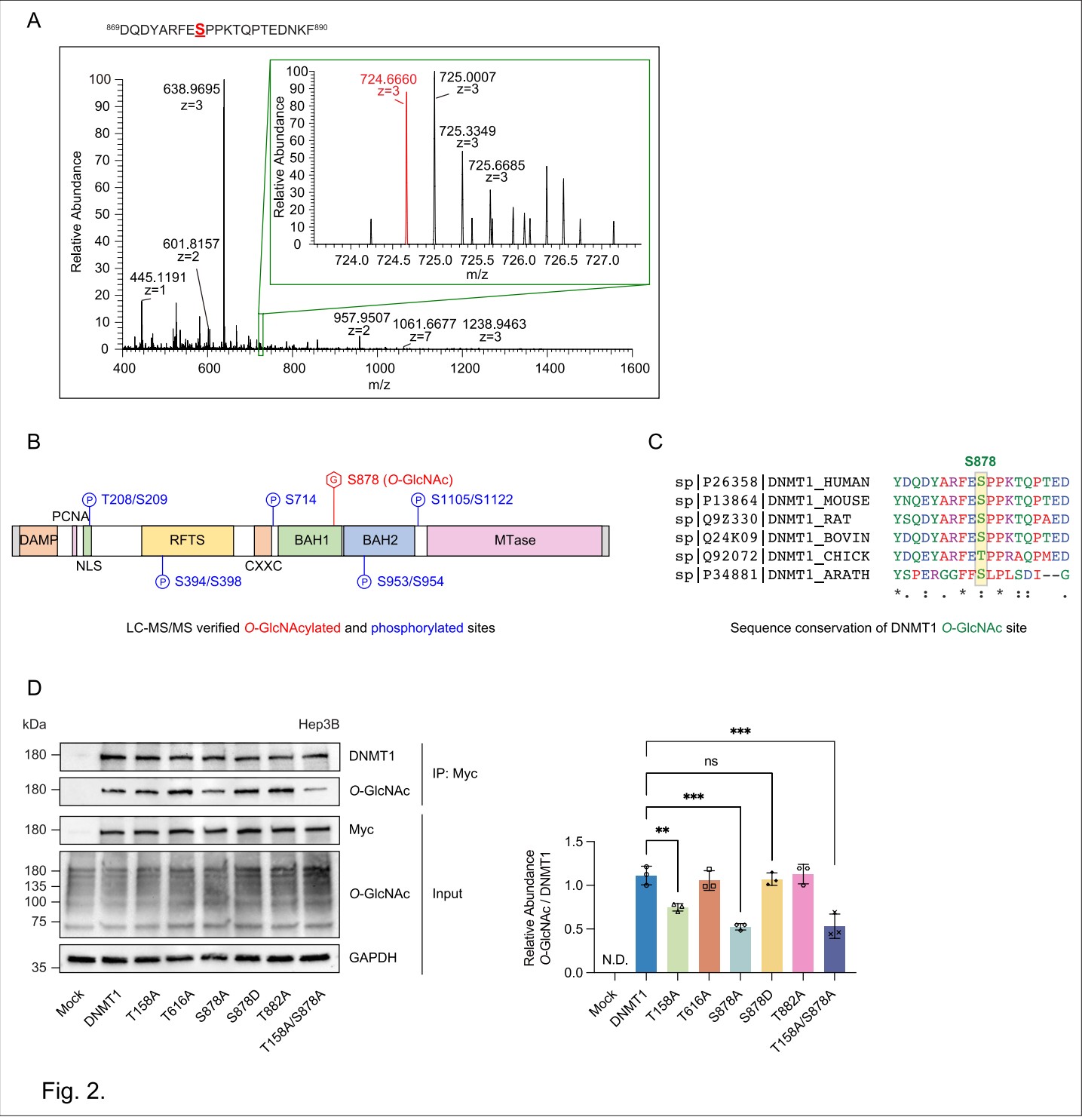

**Figure 2.** Identification of *O*-GlcNAcylated sites within DNMT1 by LC-MS/MS. (**A**) Schematic drawing of the DNMT1 *O*-GlcNAc-modified region enriched from Hep3B cells based on mass spectrometry (MS) data and tandem MS (MS/MS) peaks. FTMS+ p NSI full MS (400.0000–1600.0000). DQDYARFESPPKTQPTEDNKF (S9 HexNAc) – S878. (**B**) Schematic diagram of identified novel *O*-GlcNAcylated and phosphorylated sites within DNMT1 as determined via LC-MS/MS. DMAP, DNA methyltransferase associated protein-binding domain; PCNA, proliferating cell nuclear antigen-binding domain; NLS, nuclear localization sequences; RFTS, replication foci targeting sequence domain; BAH, bromo-adjacent homology domain. (**C**) Sequence conservation of S878 in vertebrates. (**D**) Each immunoprecipitated Myc-DNMT1 wild type and substituted mutants was immunoblotted with an *O*-GlcNAc antibody (n = 3). **p<0.0005; ***p<0.0001 by Student's *t*-test (**D**); N.D., not detected, ns, not significant; data are represented as mean ± SD from three replicates of each sample.

*Figure 2 continued on next page*

*Figure 2 continued*

The online version of this article includes the following source data and figure supplement(s) for figure 2:

**Source data 1.** Uncropped blot files of *Figure 2D*.

**Figure supplement 1.** Myc-DNMT1-WT in Hep3B cells can be *O*-GlcNAcylated.

**Figure supplement 1—source data 1.** Uncropped blot files of *Figure 2—figure supplement 1*.

**Figure supplement 2.** Tandem MS/MS peaks of *O*-GlcNAcylated DNMT1 peptides.

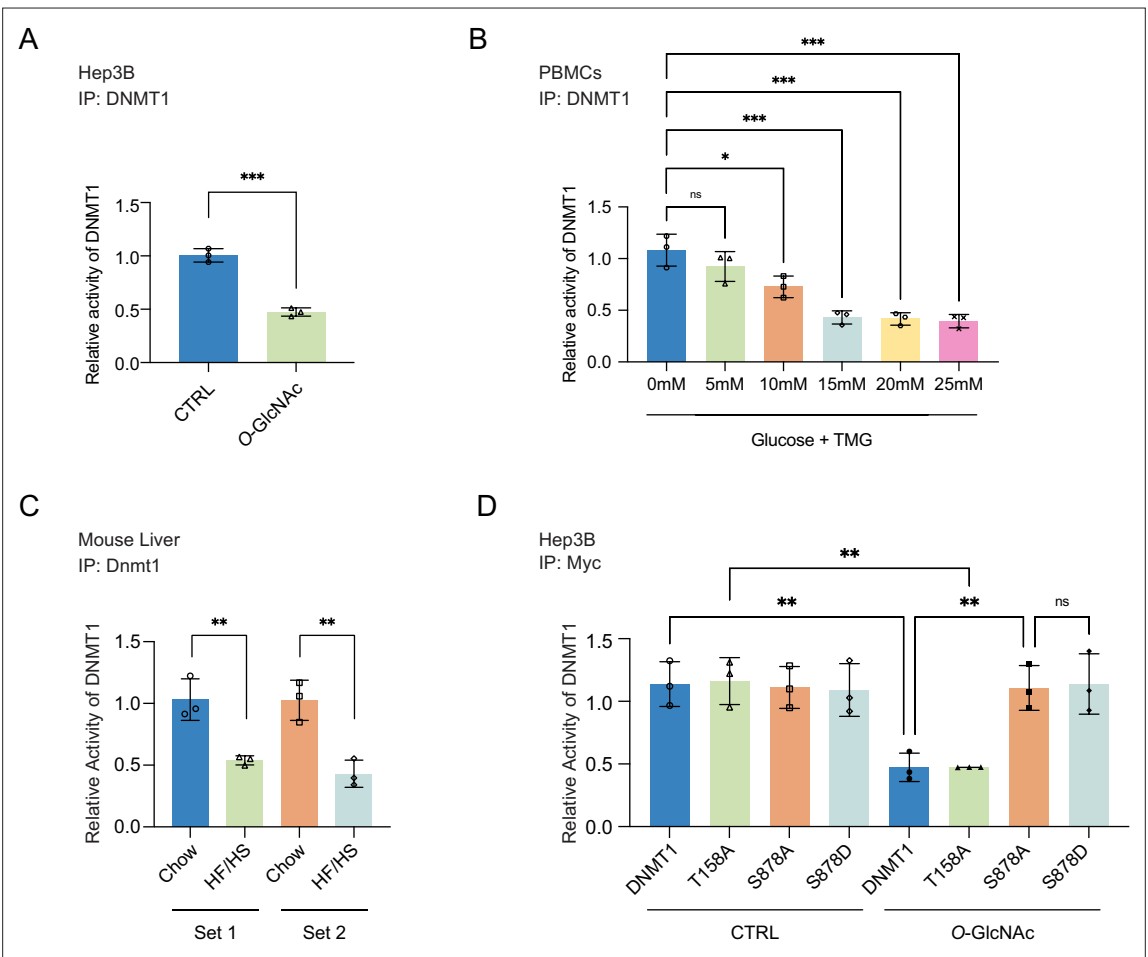

**Figure 3.** Site-specific *O*-GlcNAcylation inhibits DNMT1 methyltransferase function. For (**A–D**), bar graphs are of relative activity of DNA methyltransferase activity measured as absorbance from a DNMT activity/Inhibition ELISA kit and representative immunoblots of immunoprecipitates performed with antibodies targeting DNMT1. (**A**) Hep3B cells were treated with low (5 mM, CTRL) or high glucose/Thiamet-G (TMG) (25 mM, *O*-GlcNAc) (n = 3). (**B**) Peripheral blood mononuclear cells (PBMCs) from donors were treated with increasing concentrations of glucose (range: 0–25 mM with TMG) (n = 3). (**C**) Liver samples from C57BL/6J mice given a high-fat/high-sucrose diet (HF/HS) or a normal diet (chow) for 4 mo. (**D**) Immunoprecipitated DNMT1 wild type and substituted mutants were treated with 5 mM or 25 mM glucose (n = 3). *p<0.001; **p<0.0005; ***p<0.0001 by Student's *t*-test (**A–D**); ns, not significant; data are represented as mean ± SD from three replicates of each sample.

The online version of this article includes the following source data and figure supplement(s) for figure 3:

**Figure supplement 1.** Site-specific *O*-GlcNAcylation at DNMT1 sites abrogate the function of methyltransferase and DNA loss of methylation at CpG island under high glucose/Thiamet-G (TMG) conditions.

**Figure supplement 2.** The methylation loss by high glucose/Thiamet-G (TMG) conditions was not apparent in the DNMT1-S878A mutant.

**Figure supplement 3.** High glucose/Thiamet-G (TMG) conditions do not impact localization of DNMT1-WT or DNMT1-S878A.

**Figure supplement 3—source data 1.** Raw fluorescence image files of *Figure 3*.

We next examined the impact of glucose levels on the function of DNMT1 in primary cells by treating PBMCs with increasing concentrations of glucose (0 mM, 5 mM, 10 mM, 15 mM, 20 mM, and 25 mM with TMG) for 96 hr and measuring the DNA methyltransferase activity of DNMT1. We observed a striking dose-dependent inhibition of the DNA methyltransferase activity of DNMT1 (*Figure 3B*). Lastly, we examined the activity of DNMT1 in the liver samples of mice fed an HF/HS diet, which showed a decreased activity of DNMT1 (*Figure 3C*) compared to chow-fed mice. Together, these data indicate that elevated levels of extracellular glucose can inhibit the methyltransferase function of DNMT1.

We next examined the ability of the DNMT1 alanine mutants (DNMT1-T158A and DNMT1-S878A), which cannot be *O*-GlcNAcylated, to attenuate the impact of high glucose- and TMG-induced loss of DNA methyltransferase activity. Compared to the DNA methyltransferase activity of DNMT1-WT (Myc-DNMT1-WT), the DNA methyltransferase activity of DNMT1-S878A (Myc-DNMT1-S878A) is not inhibited by high glucose/TMG treatment (*Figure 3D*), indicating that *O*-GlcNAcylation of DNMT1-S878 is directly involved in the inhibition of methyltransferase activity. In contrast, the DNA methyltransferase activity of DNMT1-T158A (Myc-DNMT1-T158A) is inhibited by high glucose/TMG treatment in a manner similar to DNMT1-WT (Myc-DNMT1-WT), indicating that *O*-GlcNAcylation of DNMT1-T158 does not affect its DNA methyltransferase activity (*Figure 3D*).

A previous phospho-proteomic analysis revealed that DNMT1-S878 can be phosphorylated (*Zhou et al., 2013*), but the functional consequences of this have not been investigated. To evaluate the potential that phosphorylation, rather than *O*-GlcNAcylation, of S878 is leading to the loss of DNA methyltransferase activity observed with the DNMT1-S878A mutant, we generated DNMT1-S878D mutant, a phosphomimetic mutant that cannot be *O*-GlcNAcylated and examined DNA methyltransferase activity in normal and high glucose/TMG conditions. With treatment this phospho-mimetic mutant did not have loss of DNA methyltransferase activity, indicating that *O*-GlcNAcylation of S878 but not phosphorylation of S878 is leading to loss of methyltransferase activity of DNMT1 (*Figure 3D*).

## *O*-GlcNAcylation of DNMT1 results in subsequent loss of DNA methylation

Given our observations that *O*-GlcNAcylation of DNMT1 inhibits its DNA methyltransferase activity, we reasoned that this would further result in a general loss of DNA methylation. To begin to assess this, DNA methylation was assayed using the global DNA methylation LINE-1 kit (Active Motif, details in 'Materials and methods') as a proxy for global methylation. Comparison of DNA methylation under high glucose/TMG and treatment with DNA methylation inhibitors (5-aza or GSK-3484862 [*Azevedo Portilho et al., 2021*]; details in 'Materials and methods') revealed that treatment leads to a loss of DNA methylation in a manner comparable with a DNA methylation inhibitor, GSK-3484862 (*Figure 3—figure supplement 2*). The methylation loss resulting from treatment with high glucose/TMG was not apparent in the DNMT1-S878A mutant, further demonstrating that *O*-GlcNAcylation of S878 within DNMT1 directly affects DNA methylation activity (*Figure 3—figure supplement 2*). When we assessed methylation with the phosphor-mimetic mutant (DNMT1-S878D), there was no correlation between phosphorylation of DNMT1 S878 and DNA loss of methylation loss under high glucose/TMG conditions (*Figure 3—figure supplement 2*). A complementary assessment of DNA methylation using methylation-sensitive restriction enzymes and gel electrophoresis (details in 'Materials and methods') revealed similar trends of DNA methylation loss (*Figure 3—figure supplement 2*) albeit to a lesser extent using HpaII digestion (vs. MseI in the LINE-1 assay kit). The differences can also be due to the variation between the two assays in enzyme activity/digestion and differences in hybridization vs. gel separation. To evaluate alterations to DNA methylation more precisely, we utilized direct sequencing of genomic DNA, which revealed an overall loss of DNA methylation across the genome.

To examine whether the genomic loss of DNA methylation under high glucose conditions is affected by changes in DNMT1 localization in the cell (*Jones et al., 2021*), we examined changes in the cellular localization of DNMT1 by immunofluorescence. However, changes in DNMT1 under hyperglycemia do not appear to impact its nucleocytoplasmic localization (*Figure 3—figure supplement 3*).

## O-GlcNAcylation of DNMT1 results in loss of DNA methylation at PMDs

To more thoroughly examine the impact of high glucose-induced O-GlcNAcylation of DNMT1 on the epigenome, Myc-DNMT1-overexpressed cell lines (DNMT1-WT and DNMT1-S878A) were treated with either low (5 mM, CTRL) or high glucose/TMG (25 mM, O-GlcNAc) and DNA methylation was profiled with nanopore sequencing (ONT PromethION; details in 'Materials and methods'). A comparison of the methylation profiles in these cells revealed a global loss of methylation in high glucose compared to control (*Figure 4A and B* and *Figure 4—figure supplement 1*). Conversely, for the DNMT1-S878A mutant, there was no appreciable decrease in DNA methylation by high glucose (*Figure 4A*). These results collectively indicate that O-GlcNAcylation of S878 of DNMT1 leads to a global loss of DNA methylation.

Examination of DNA methylation changes induced by O-GlcNAcylation of DNMT1 revealed a preferential loss of DNA methylation at liver cancer PMDs (*Li et al., 2016*) that was not observed in S878A mutant cells (*Figure 4B and C*). PMDs have several defining features, including being relatively gene poor and harboring mostly lowly transcribed genes (*Decato et al., 2020*). We stratified the genome in terms of gene density and transcription rate (see 'Materials and methods' for details) and found that regions that lose methylation in high glucose conditions are largely gene-poor (*Figure 4D*) and contain lowly transcribed genes (*Figure 4—figure supplement 2*; *Chang et al., 2014*). PMDs have furthermore been linked to regions of late replication associated with the nuclear lamina (*Brinkman et al., 2019*). We therefore examined the correlation between loss of methylation caused by high glucose and replication timing (*Thurman et al., 2007*). In DNMT1-WT cells, late replication domains preferentially lose DNA methylation in high glucose/TMG conditions compared to early replication domains (*Figure 4E*). This loss of methylation was not observed in DNMT1-S878A mutant cells (*Figure 4—figure supplement 3*).

## Evolutionarily young TEs are protected from loss of methylation in high glucose conditions

One of the major functions of DNA methylation in mammalian genomes is the repression of repetitive elements (*Edwards et al., 2017*). Furthermore, it has been shown that many chromatin proteins involved in the repression of TEs are capable of being O-GlcNAcylated (*Boulard et al., 2020*). We therefore examined the potential of O-GlcNAcylation of DNMT1 to lead to loss of suppression of TEs. We found that high glucose conditions resulted in methylation loss at TEs in a manner similar to the nonrepetitive fraction of the genome (*Figure 4—figure supplement 4*), with a more dramatic loss of methylation at LINEs and LTRs compared to SINE elements (*Figure 4—figure supplement 4*). Given that evolutionarily recent TEs are more likely to lose methylation than older elements in a variety of systems (*Almeida et al., 2022*; *Zhou et al., 2020*), we examined the methylation status of two younger subfamilies, LTR12C (*Hominoidea*) and HERVH-int (*Catarrhini*) elements (*Figure 4—figure supplement 5*). While HERVH-int elements show a loss of methylation similar to the rest of the genome, LTR12C elements do not lose methylation in the same manner (*Figure 4—figure supplement 5*), suggesting that they are protected from the loss of methylation. To identify the possible regulatory mechanisms behind the continued maintenance of methylation of LTR12C elements, we examined binding data (ChIP-exo) of KRAB-associated zinc-finger proteins (*Imbeault et al., 2017*), a family of proteins associated with the regulation of transposons, on all LTR12Cs present within hepatic cancer PMDs (*Li et al., 2016*). ZFP57 and ZNF605 demonstrate binding to a significant number of LTR12C elements present in liver cancer PMDs (*Figure 4—figure supplement 6*), suggesting that binding of these KZFPs could maintain the methylation at LTR12Cs. Stratifying all TEs by evolutionary age (*Storer et al., 2021*) and examining the methylation changes induced by O-GlcNAcylation of DNMT1 for each clade revealed that evolutionarily recent elements in general are less likely to lose methylation than older elements (*Figure 4—figure supplement 7*).

## Methylation changes at promoter regions of apoptosis and oxidative stress response genes upon inhibition of DNMT1

To further examine the impact of the altered epigenome in high glucose conditions, we examined the methylation levels of promoter regions (defined as the 2 kb window upstream and downstream of the transcription start site [TSS]) (*Figure 4—figure supplement 1* and *Figure 5—figure supplement*

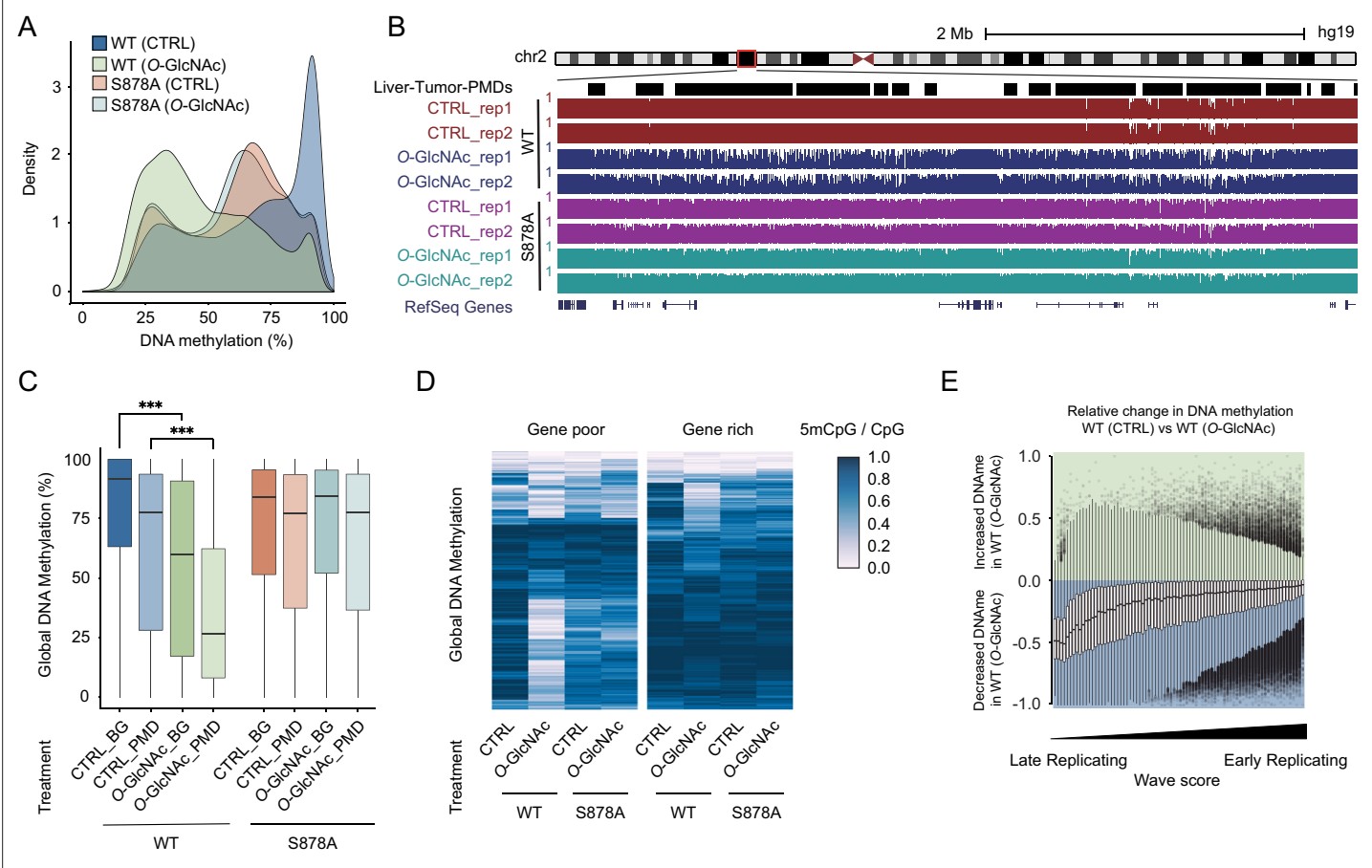

**Figure 4.** High glucose leads to loss of DNA methylation at cancer-specific partially methylated domains (PMDs). (**A**) Density plot of DNA methylation for DNMT1-WT and DNMT1-S878A cells with either low (5 mM, CTRL) or high glucose/Thiamet-G (TMG) (25 mM, *O*-GlcNAc). (**B**) Genome browser screenshot of DNA methylation for DNMT1-WT and DNMT1-S878A cells and low or high glucose along with liver tumor PMDs from *Li et al., 2016*. (**C**) Boxplots of DNA methylation at PMDs or general genomic background (BG) for each DNMT1-WT and DNMT1-S878A treated with low (5 mM, CTRL) or high glucose/TMG (25 mM, *O*-GlcNAc). (**D**) Heatmap representation of global DNA methylation for DNMT1-WT and DNMT1-S878A cells under low (5 mM, CTRL) or high glucose/TMG (25 mM, *O*-GlcNAc) at gene-poor and gene-rich regions. (**E**) Methylation changes from *O*-GlcNAcylation of DNMT1 by wave score for replication timing (*Hansen et al., 2010*; *Thurman et al., 2007*). ***p<0.0001 by Wilcoxon signed-rank test (**C**).

The online version of this article includes the following figure supplement(s) for figure 4:

**Figure supplement 1.** DNA loss of methylation by increased global *O*-GlcNAcylation decreases.

**Figure supplement 2.** Global DNA methylation of wild type and DNMT1 mutants between low Fragments Per Kilobase of transcript per Million mapped reads (FPKM) regions and high FPKM regions (DNMT1-WT or DNMT1-S878A) that treated low (5 mM, CTRL) or high glucose/Thiamet-G (TMG) (25 mM, *O*-GlcNAc) were determined by Nanopolish call methylation.

**Figure supplement 3.** Methylation changes from *O*-GlcNAcylation of DNMT1 in DNMT1-S878A mutant.

**Figure supplement 4.** DNA loss of methylation by increased global *O*-GlcNAcylation decreases around the transposable element (TE) regions.

**Figure supplement 5.** Loss of DNA methylation by increased global O-GlcNAcylation decreases.

**Figure supplement 6.** ZFP57 and ZNF605 demonstrate binding to a significant number of LTR12C elements present in liver cancer partially methylated domains (PMDs).

**Figure supplement 7.** Evolutionarily recent elements are less likely to lose methylation induced by *O*-GlcNAcylation of DNMT1.

**Figure supplement 8.** Only loci with >5× coverage were retained for analysis, comprising 90% of CpGs in the genome.

*1*). Hypermethylated (gain of methylation) and hypomethylated (loss of methylation) genes were classified as genes with differentially methylated regions overlapping the promoters (*Figure 5—figure supplement 1*). Pathway analysis (see 'Materials and methods' for details) revealed that genes with hypomethylated promoters are involved in apoptosis and oxidative stress response pathways (*Figure 5—figure supplement 2*). Examination of apoptosis-related proteins using an apoptosis

proteome array revealed that apoptosis agonist proteins (cleaved-caspase3 and phopho-p53 [S15]) are increased and antagonistic proteins (pro-caspase3, surviving, and claspin) are decreased by high glucose treatment (*Figure 5—figure supplement 3*). Intriguingly, only cIAP-1 is increased in the high glucose-treated DNMT1-S878A mutant cells (*Figure 5—figure supplement 3*).

## DNA hypomethylation induced DNA damage and triggers apoptosis by high glucose

High glucose-induced generation of reactive oxygen species (ROS) has been shown to result in increased cell death (*Allen et al., 2005*). Increased ROS has furthermore been shown to result in upregulation of DNMT1 (*He et al., 2012*; *O'Hagan et al., 2011*). To further explore the link between glucose levels, DNMT1, and cell death, we treated DNMT1-WT and DNMT1-S878A cells with either low or high glucose/TMG for 96 hr and examined the fluorescence of 2',7'-dichlorofluorescein diacetate (DCFH-DA) as an indicator for ROS (*Figure 5A*). The levels of ROS were increased upon treatment with 25 mM glucose with TMG in both DNMT1-WT and DNMT1-S878A (*Figure 5A*). Given that high levels of ROS can lead to increased DNA damage and subsequent cell death (*Rowe et al., 2008*), we analyzed DNA damage using quantitative fluorescence image of γ-H2A.X in DNMT1-WT and DNMT1-S878A cells and low (5 mM, CTRL) or high glucose/TMG (25 mM, *O*-GlcNAc). Interestingly, DNA damage indicated by levels of γ-H2A.X was reduced in high glucose/TMG-treated DNMT1-S878A cells compared to WT cells despite high ROS levels (*Figure 5B*). To further confirm that this damage is from oxidative stress, we analyzed 8-hydroxy-2'deoxyguanosine (8-OHdG) as a marker for oxidative DNA damage using EpiQuik 8-OhdG DNA damage quantification kit (EpiGentek, details in 'Materials and mmethods'). This also reduced DNA oxidative damage in high glucose-treated DNMT1-S878A cells compared to WT cells (*Figure 5—figure supplement 4*). Furthermore, DNA damage induced by low glucose with GSK-3484862 treatment suggests that DNA hypomethylation is associated with increased DNA damage (*Figure 5—figure supplement 4*; *Azevedo Portilho et al., 2021*). Finally, examination of propidium iodide (PI) levels revealed that cell death was prominently increased in the high glucose-treated DNMT1-WT cells but suppressed in the DNMT1-S878A mutants (*Figure 5C*). Taken together, these results suggest that ROS-induced DNA damage under hyperglycemic conditions is partially mitigated by DNA methylation; elevated *O*-GlcNAcylation of DNMT1 coinciding with DNA methylation loss is associated with more ROS-induced DNA damage and increased cell death. These results indicate that extracellular metabolic stress and cell fate are linked through epigenetic regulation.

## Discussion

Although there is a great deal of evidence regarding the important regulatory role of *O*-GlcNAcylation in gene regulation (*Brimble et al., 2010*), a direct link with maintenance of DNA methylation has not previously been established. The maintenance methyltransferase DNMT1 is essential for faithful maintenance of genomic methylation patterns and mutations in DNMT1, particularly in the BAH domains, lead to disruption of DNA methylation (*Gong et al., 2021*; *Yarychkivska et al., 2018*). The function of the Bromo-Adjacent Homology (BAH) domain is to mediate association of DNMT1 with replication foci during S phase to maintain methylation. The BAH-deleted form of DNMT1 was unable to maintain DNA methylation genome-wide or in IAP retrotransposons and major satellite DNA, commonly densely methylated sequences (*Yarychkivska et al., 2018*). In addition, the DNMT1 S878F mutation results in hypomethylation of the HBG promoter region (*Gong et al., 2021*). While it has been shown that DNMT1 can be *O*-GlcNAcylated (*Boulard et al., 2020*), the site of *O*-GlcNAc modification on DNMT1 – as well as the functional consequences of this modification – has not previously been examined. In this study, we detected *O*-GlcNAcylation in the BAH1 domain of DNMT1 under high glucose/TMG conditions in cultured cells lines as well as in primary human and mouse cells. In addition, it was confirmed that the DNA methyltransferase activity of DNMT1 was directly inhibited by *O*-GlcNAcylation. Our findings suggest that the direct *O*-GlcNAcylation of DNMT1 disrupts the BAH1 domain, thereby suppressing the function of DNMT1. Therefore, this study provides novel insights into the mechanism of *O*-GlcNAcylation-mediated regulation of DNMT1 and global DNA methylation. Further studies are needed to clarify the molecular mechanism by which *O*-GlcNAcylation disrupts the BAH1 domains and regulates DNMT1 and DNA methylation.

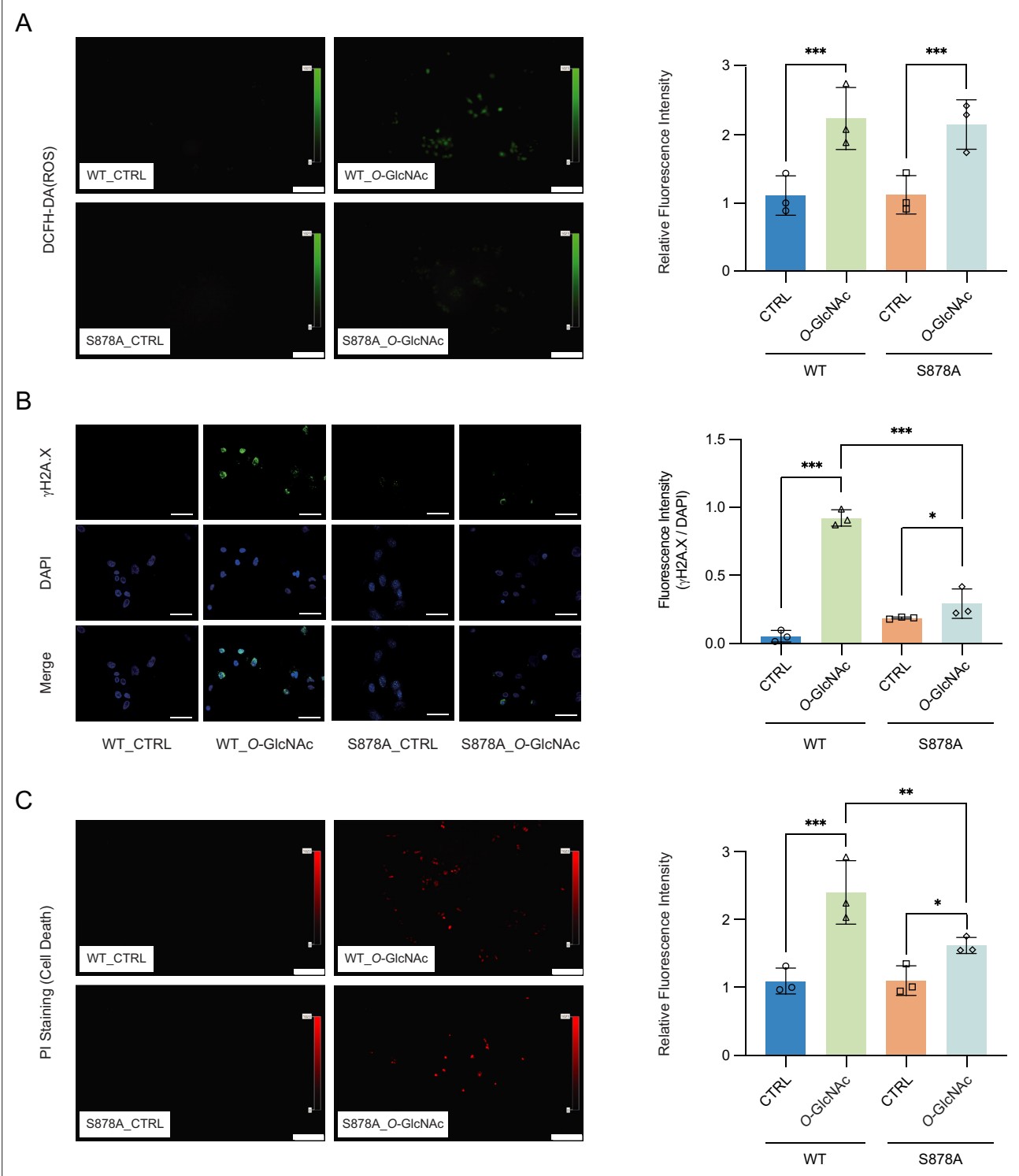

**Figure 5.** High glucose-induced reactive oxygen species (ROS) and DNA damage cause apoptotic cell death in DNMT1-WT cells. (**A**) Quantitative fluorescence image of ROS in DNMT1-WT and DNMT1-S878A cells with either low (5 mM, CTRL) or high glucose/Thiamet-G (TMG) (25 mM, *O*-GlcNAc). (**B**) Quantitative fluorescence image of γ-H2A.X in DNMT1-WT and DNMT1-S878A cells treated with low (5 mM, CTRL) or high glucose/TMG (25 mM, *O*-GlcNAc). (**C**) Quantitative fluorescence image of cell death in propidium iodide staining of DNMT1-WT and DNMT1-S878A cells under low (5 mM, CTRL) or high glucose/TMG (25 mM, *O*-GlcNAc). *p<0.001; **p<0.0005; ***p<0.0001 by Student's *t*-test (**A–C**); data are represented as mean ± SD from three replicates of each sample.

The online version of this article includes the following source data and figure supplement(s) for figure 5:

*Figure 5 continued on next page*

*Figure 5 continued*

**Source data 1.** Raw fluorescence image files of *Figure 5A–C*.

**Figure supplement 1.** Heatmap representation of promoter DNA methylation for DNMT1-WT and DNMT1-S878A cells under low (5 mM, CTRL) or high glucose/Thiamet-G (TMG) (25 mM, *O*-GlcNAc) at gene-poor and gene-rich regions.

**Figure supplement 2.** DNA loss of methylation within promoter region by increased global *O*-GlcNAcylation impact different gene pathways.

**Figure supplement 3.** Quantitative analysis of human apoptosis related proteins in DNMT1-WT and DNMT1-S878A with either low (5 mM, CTRL) or high glucose/Thiamet-G (TMG) (25 mM, *O*-GlcNAc) using Proteome profiler (n = 3).

**Figure supplement 3—source data 1.** Uncropped blot files of *Figure 5—figure supplement 3*.

**Figure supplement 4.** Each Hep3B (Mock) and Myc-DNMT1-overexpressed mutant (DNMT1-WT or DNMT1-S878A) was treated with low (5 mM, CTRL) or high glucose/Thiamet-G (TMG) (25 mM, *O*-GlcNAc), 5-aza, or GSK-3484862 (negative control).

Our results reveal that *O*-GlcNAcylation of DNMT1 impacts its DNA methyltransferase activity and affects DNMT1 function, leading to loss of DNA methylation at PMDs. PMDs have been observed in both healthy and disease cells and have been suggested to be associated with mitotic dysfunction. However, models for how these domains are established remain incomplete (*Decato et al., 2020*). The results presented here suggest an additional layer whereby *O*-GlcNAcylation of DNMT1 at S878 due to increased glucose levels can inhibit the function of DNA methyltransferase activity of DNMT1, resulting in loss of methylation and establishment of PMDs.

High glucose conditions have previously been reported to lead to an increase in nuclear 25-hydroxycholesterol, which induces lipid accumulation and activates DNMT1 (*Allen et al., 2005*; *Wang et al., 2020*). Our results are consistent with the activity of DNMT1 gradually increasing with glucose concentrations (*Figure 3—figure supplement 1*). This trend is reversed, however, upon TMG treatment (*Figure 3—figure supplement 1*), suggesting that the increased activity of DNMT1 associated with glucose treatment is directly inhibited by *O*-GlcNAcylation within DNMT1.

Protein *O*-GlcNAcylation has recently been explored in the context of various diseases, including cancer, diabetes, and neurological diseases. Dysregulation of *O*-GlcNAcylation, driven by increased blood glucose levels in diabetes (*Ling and Rönn, 2019*) or the Warburg effect in cancer cells, has been shown to play a crucial role in disease progression (*Liberti and Locasale, 2016*; *Vander Heiden et al., 2009*), which results in an increase in intracellular protein *O*-GlcNAcylation (*Hart et al., 2007*; *Hart et al., 1996*). *O*-GlcNAcylation acts as a molecular switch that can modulate protein function in response to nutrient changes and serve as a molecular sensor for specific cellular signals (*Vaidyanathan et al., 2014*; *Yang et al., 2017a*). Perturbations in the activity of *O*-GlcNAc transferase (OGT) and *O*-GlcNAcase (OGA) enzymes (*Wells et al., 2001*; *Zhang et al., 2003*), as well as alterations in posttranslational modifications, can disrupt protein functions vital for cell survival and contribute to metabolic reprogramming (*Shin et al., 2018*; *Yang et al., 2006*). The increased levels of *O*-GlcNAcylation observed in most diabetic patients and tumors (*de Queiroz et al., 2014*) suggest its potential utility as a biomarker. By comparing the degree of protein glycosylation in specific diseases, such as diabetes or tumors (*Lu et al., 2022*), with that in healthy cells, *O*-GlcNAcylation could potentially serve as a valuable diagnostic tool. Moreover, targeting *O*-GlcNAc-modified proteins with antibody-based therapeutics holds promise for cancer treatment and suppression of diabetic complications. By inducing tumor cell necrosis or inhibiting the abnormal activity associated with diabetic complications, these approaches represent innovative avenues for disease management. Our study highlights the impact of *O*-GlcNAcylation on DNA methyltransferase 1 (DNMT1), revealing its role in modulating DNA methylation. *O*-GlcNAcylation of DNMT1 impairs its methyltransferase function, leading to global loss of DNA methylation and aberrant activity of transposable elements. Targeting this aberrant modification with antibody therapeutics presents an exciting opportunity to restore normal DNA methylation patterns and mitigate disease complications. Continued research in this area will unveil the full potential of *O*-GlcNAcylation as a versatile tool in disease diagnosis and therapeutic interventions.

Metabolic diseases such as obesity and diabetes have been linked to epigenetic changes that alter gene regulation (*Ling and Rönn, 2019*). It has previously been established that there is a general increase in protein *O*-GlcNAcylation in hyperglycemia conditions (*Vasconcelos-Dos-Santos et al., 2018*) and several epigenetic regulatory factors have been shown to have increased *O*-GlcNAcylation under high glucose conditions (*Bauer et al., 2015*; *Etchegaray and Mostoslavsky, 2016*; *Yang et al.,*

2020). Our findings that extracellular glucose promotes *O*-GlcNAcylation of DNMT1 and inhibition of DNMT1's function in maintenance of genomic methylation provide direct evidence that extracellular levels of glucose impacts epigenomic regulation.

# Materials and methods

## Key resources table

| Reagent type (species) or resource | Designation | Source or reference | Identifiers | Additional information |
|---|---|---|---|---|
| Gene (*Homo sapiens*) | *DNMT1* | HUGO Gene Nomenclature Committee | HGNC:2976 | - |
| Cell line (*H. sapiens*) | Hep 3B2.1–7 | ATCC | HB-8064 | - |
| Cell line (*H. sapiens*) | Hep G2 | ATCC | HB-8065 | - |
| Transfected construct (*H. sapiens*) | pcDNA3/Myc-DNMT1 | Addgene | Plasmid #36939 | |
| Antibody | Anti-beta-actin (D6A8) (rabbit monoclonal) | Cell Signaling Technology | Cat# 8457 | WB (1:1000) |
| Antibody | Anti-alpha-tubulin (11H10) (rabbit monoclonal) | Cell Signaling Technology | Cat# 2125 | WB (1:1000) |
| Antibody | Anti-DNMT1 (60B1220.1) (mouse monoclonal) | Novus Biologicals | Cat# NB100-56519 | IP (1:250) WB (1:1000) |
| Antibody | Anti-DNMT1 (H-12) (mouse monoclonal) | Santa Cruz Biotechnology | Cat# sc-271729 | WB (1:1000) |
| Antibody | Anti-gamma H2A.X (rabbit polyclonal) | Abcam | Cat# ab11174 | IF (1:1000) |
| Antibody | Anti-GAPDH (rabbit monoclonal) | Abcam | Cat# ab181602 | WB (1:1000) |
| Antibody | Anti-H3 (rabbit polyclonal) | Abcam | Cat# ab1791 | WB (1:1000) |
| Antibody | Anti-Myc [Myc.A7] (mouse monoclonal) | Abcam | Cat# ab18185 | IP (1:250) WB (1:1000) |
| Antibody | Anti-*O*-GlcNAc (RL2) (mouse monoclonal) | Abcam | Cat# ab2739 | WB (1:1000) |
| Antibody | Aoat anti-rabbit IgG H&L (HRP) (goat polyclonal) | Abcam | Cat# ab6721 | WB (1:5000) |
| Antibody | Goat anti-mouse IgG H&L (HRP) (goat polyclonal) | Abcam | Cat# ab6789 | WB (1:5000) |
| Antibody | Goat anti-mouse IgG (H+L), Alexa 488 (goat polyclonal) | Invitrogen | Cat# A32723 | WB (1:1000) |
| Sequence-based reagent | DNMT1-T158A | This paper | PCR primers | agccccaggatt CGA aggaaaagcacc |
| Sequence-based reagent | DNMT1-T158A | This paper | PCR primers | ggtgcttttcct TCG aatcctggggct |
| Sequence-based reagent | DNMT1-T616A | This paper | PCR primers | gacaggggaccc GCG aaagccaccacc |
| Sequence-based reagent | DNMT1-T616A | This paper | PCR primers | ggtggtggcttt CGC gggtcccctgtc |
| Sequence-based reagent | DNMT1-S878A | This paper | PCR primers | gcgagattcgag GAG cctccaaaaacc |
| Sequence-based reagent | DNMT1-S878A | This paper | PCR primers | ggtttttggagg CTC ctcgaatctcgc |
| Sequence-based reagent | DNMT1-S878D | This paper | PCR primers | gcgagattcgag GAC cctccaaaaacc |
| Sequence-based reagent | DNMT1-S878D | This paper | PCR primers | ggtttttggagg GTC ctcgaatctcgc |
| Sequence-based reagent | DNMT1-T882A | This paper | PCR primers | tcccctccaaaa GCC cagccaacagag |
| Sequence-based reagent | DNMT1-T882A | This paper | PCR primers | ctctgttggctg GGC ttttggaggga |
| Commercial assay or kit | Q5 Site-Directed Mutagenesis kit | NEB | Cat# E0554S | - |

*Continued on next page*

*Continued*

| Reagent type (species) or resource | Designation | Source or reference | Identifiers | Additional information |
|---|---|---|---|---|
| Commercial assay or kit | EpiQuik DNMT Activity/Inhibition ELISA Easy Kit | EpiGentek | Cat# P-3139 | - |
| Commercial assay or kit | Global DNA methylation LINE-1 | Active Motif | Cat# 55017 | - |
| Commercial assay or kit | EpiQuik 8-OHdG DNA Damage Quantification Direct kit | EpiGentek | Cat# P-6003 | - |
| Chemical compound, drug | OSMI-4 | Selleck Chem | Cat# S8910 | - |
| Chemical compound, drug | Thiamet-G (TMG) | Cayman Chemical | Cat# 13237 | - |
| Software, algorithm | bedGraphToBigWig | *Kent et al., 2010* | - | - |
| Software, algorithm | Clustal Omega | *Sievers et al., 2011* | - | Version 1.2.4 |
| Software, algorithm | GraphPad Prism 9 | GraphPad | - | Version: 9.3.1 |
| Software, algorithm | Minimap2 | *Li and Birol, 2018* | RRID:SCR_018550 | Version: 2.17 |
| Software, algorithm | Nanopolish | *Loman et al., 2015* | RRID:SCR_016157 | Version: 0.11.1 |
| Software, algorithm | Python | Python Core Team | | Version: 3.8.2 |
| Software, algorithm | R | R Core Team | - | Version: 3.4.3 |
| Software, algorithm | Samtools | *Lister et al., 2009* | RRID:SCR_002105 | Version: 1.10 |
| Other | UniProt | The UniProt Consortium | - | Database of protein information (https://www.uniprot.org/) |
| Other | DAPI stain | Invitrogen | D1306 | 1 µg/ml; marker for nuclear DNA |

## Cell culture and plasmid DNA transfection

Human hepatocellular carcinoma cell lines Hep3B (HB-8064) and HepG2 (HB-8065) were purchased from ATCC (Manassas, VA). These cells were authenticated by ATCC with STR Profiling Results. All cell lines were shown to be negative in mycoplasma test using MycoScope (MY01050, Genlantis, San Diego, CA). The following ATCC-specified cell culture media were used: Dulbecco's modified Eagle's medium (DMEM, 11885-084, Gibco, Grand Island, NY) or high glucose DMEM (11995-065, Gibco) with 10% fetal bovine serum (FBS, SH30910.03, HyClone, South Logan, UT) and Opti-MEM (1869048, Gibco). All cells were cultured at 37°C with a 5% $CO_2$ atmosphere incubator. HepG2 and Hep3B cells were transiently transfected (with the pcDNA3 with or without DNMT1 cDNA) using Lipofectamine 3000 (Invitrogen, Carlsbad, CA) and selected with Geneticin (G418, 10131-035, Gibco) according to the manufacturer's instructions. Human DNMT1 plasmid was purchased from Addgene (#36939, Watertown, MA; *Li et al., 2006*).

## Glucose-stress experimental conditions

Cell culture medium (DMEM) was replaced every 24 hr with low or high glucose. The glucose stress durations of the experiment were either 24 hr or 96 hr. Low glucose medium contains 5 mM glucose (normal physiological glucose level), whereas high glucose medium contains 25 mM glucose (hyperglycemic range >7 mM) and corresponds to severe diabetes (*Koobotse et al., 2020*). Thiamet-G (#13237, Cayman Chemical, Ann Arbor, MI) was added directly to the cell culture medium.

## Isolation of PBMCs

Blood samples from deidentified healthy donors were obtained following guidelines at the City of Hope as previously described (*Leung et al., 2018*). PBMCs were isolated directly from human whole blood using Ficoll-Paque (Premium, GE Healthcare, Chicago, IL) density gradient centrifugation. 15 ml whole blood was mixed with same volume of phosphate-buffered saline containing 0.1% FBS +2 mM EDTA (PBS solution). Next, the blood mix was placed on top of 15 ml Ficoll and centrifuged at 400 × $g$ to 200 × $g$ for 40 min without brake. Next, remove the supernatant and wash three times with PBS solution.

## Isolation of B cells and Epstein–Barr virus (EBV) infection for lymphocyte transformation

CD19+ B cells were isolated from PBMCs using Dynabeads CD19+ pan B (11143D, Invitrogen) according to the manufacturer's instructions. $2.5 \times 10^8$ cells of PBMCs were resuspended in 10 ml isolation buffer (PBS, 0.1% BSA, 2 mM EDTA). 250 µl of prewashed beads were added to PBMCs and incubated for 20 min in 4°C with gentle rotation. For positive isolation of CD19+ B cells, beads and supernatant were separated using magnet, and supernatant was discarded. Beads were washed three times, and beads bounded with CD19+ B cells were resuspended with 2.5 ml of cell culture medium (80% RPMI 1640, 20% heat-inactivated FBS, glutamine). CD19+ B cells were released from Dynabeads using DETACHaBEAD (Invitrogen, ca12506D) according to the manufacturer's instruction.

B cells were infected with EBV to transform lymphocytes. 10 ml of B cells were transferred into a T75 flask. 1.5 ml of stock EBV collected from a B95-8 strain-containing marmoset cell line and 1 ml of phytohemagglutinin P (PHA-P) were added to flask and incubated in 37°C with a 5% $CO_2$ atmosphere incubator. Every 5–7 d, 10 ml of cell culture medium was added. Cells were let to grow in the $CO_2$ atmosphere incubator for 30 d until all B cells were transformed to LCLs.

## Mouse liver samples

All animal experiments conducted have been approved by the Institutional Animal Care and Use Committees at City of Hope. All of the animals were handled according to approved institutional animal care and use committee (IACUC) protocols (#17010). C57BL/6J mice were randomized to receive irradiated HF/HS diet (D12266Bi, Research Diets Inc, 17% kcal protein, 32% kcal fat, 51% kcal carbohydrate) starting at 8 weeks old for 16 wk. Mice on chow diet (D12489Bi, Research Diets Inc, 16.4% kcal protein, 70.8% kcal carbohydrate, 4.6% kcal fat) were fed for the same duration. To reduce blood contamination, mice were washed 10× with phosphate-buffered saline (PBS) solution. Washed liver tissues (two chow and two HF/HS) were cut into several pieces and divided into three groups each (three sets per each condition, total 12 samples). Each group of washed liver tissues were lysed with non-detergent IP buffer in the presence of a protease inhibitor (Cat# 8340; Sigma-Aldrich) and a phosphatase inhibitor cocktail (Cat# 5870; Cell Signaling) for the western blotting or immunoprecipitation. An increase in fasting blood glucose levels due to the HF/HS diet has been previously reported (*Franson et al., 2021*; *Tang et al., 2020*). At the end points, mice were euthanized with $CO_2$ inhalation.

## Immunoprecipitation and western blot analysis

Cell lysates were incubated with specific antibodies and lysis buffer for 4 hr. Subsequently, 30 µl of washed Dynabeads (14311D, Thermo Fisher, Waltham, MA) were added to each lysate and incubated overnight at 4°C. Next, the beads were washed five times, and the antigens were eluted twice using 8 M urea buffer (8 M urea, 20 mM Tris pH 7.5, and 100 mM NaCl) and concentrated. The resulting samples were separated by Mini-PROTEAN TGX (4–20%, 4561093, Bio-Rad Laboratories, Hercules, CA) and transferred onto nitrocellulose membranes (Amersham Hybond, 10600021, GE Healthcare) using Trans-Blot SD Semi-dry Transfer Cell system (Bio-Rad Laboratories). The membranes were then blocked with 5% skim milk in Tris-buffered saline + Tween-20 (TBS-T; 20 mM Tris, 137 mM NaCl, 0.1% Tween-20, pH 7.6), incubated overnight at 4°C with a 1:1000 dilution of each antibody, and subsequently incubated for 1 hr with a 1:5000 dilution of a horseradish peroxidase–conjugated goat anti-mouse secondary antibody (ab6789, Abcam, Cambridge, UK) or goat anti-rabbit secondary antibody (ab6721, Abcam). Immunoreactive proteins were detected using SuperSignal West Dura Extended Duration Substrate (34076, Thermo, Rockford, IL) and detected using a ChemiDoc MP Imaging system (Bio-Rad Laboratories). The band intensity was densitometrically evaluated using Image Lab software (version 5.2, Bio-Rad Laboratories).

## Protein identification using the Thermo Fusion Lumos system LC-MS/MS

Recombinant DNMT1 protein (30 mg) was reduced with dithiothreitol and alkylated with iodoacetamide. To ensure coverage of the predicted *O*-GlcNAc site, the protein was separately digested with three complementary enzymes (Chymotrypsin, AspN, and LysC). Digested peptides were desalted using $C_{18}$ stage tips (Pierce, Rockford, IL). LC-MS/MS analysis was performed on an Thermo Orbitrap

Fusion Lumos system (Thermo Scientific, San Jose, CA). Peptides were separated on an Ultimate 3000 (Thermo Scientific) operated in nanoflow (300 nl/mi) mode using a 60 min LC gradient and an EasySpray column (2 μm particle sizes: 500 mm × 75 μm I.D.). Full MS scans were acquired from 400 to 1600 *m/z* at 120,000 resolutions. MS/MS spectra were acquired at a resolution of 30,000 using HCD at 35% collision energy.

Raw data files were submitted to Byonic (v2.16.11) for target decoy search against the human protein database (*UniProt, 2023*) assuming static Cys carbamidomethylation and dynamic Met oxidation, Ser/Thr GlcNAc, and Ser/Thr phosphorylation. Only peptides passing the false discovery rate threshold 0.01 were considered for downstream analysis. The sample was bracketed by *Escherichia coli* QC runs, which were then correlated to ensure instrument quality. QC passed threshold (≥0.9) with an $R^2$ of 0.98 (correlation value, $R$ = 0.99).

## Conservation of DNMT1 sequence

The evolutionary conservation of DNMT1 was assessed via multiple sequence alignment with Clustal Omega (*Sievers et al., 2011*). DNMT1 sequences for *Homo sapiens*, *Mus musculus*, *Rattus norvegicus*, *Bos taurus*, *Gallus gallus,* and *Arabidopsis thaliana* were obtained from *UniProt, 2023* and aligned with Clustal Omega using default parameters.

## Site-directed point mutation

Specific primers for serine (S) and threonine (T) to alanine (A) and aspartic acid (D) mutations of DNMT1 were designed and used to site-directed point mutations in a plasmid vector. A recombinant DNA pcDNA3/Myc-DNMT1 was a gift from Arthur Riggs (Addgene plasmid #36939 Watertown, MA; *Li et al., 2006*). A PCR-amplified DNA fragment of pcDNA3-DNMT1 was generated using Q5 Site-Directed Mutagenesis Kit (E0554S, NEB, Ipswich, MA). The primers used in this process are described in Supporting Information. After PCR, the non-mutated sequences were cleaved using Q5 KLD enzyme (New England Biolabs, Ipswich, MA) according to the manufacturer's instructions. The mutated vectors were transformed into *E. coli* competent cells (NEB 5-alpha, New England Biolabs) that were cultured and prepared using a GenElute HP Plasmid Midi kit (NA0200-1KT, Sigma-Aldrich, St. Louis, MO).

## DNA methyltransferase activity assay

DNA methyltransferase activities of endogenous DNMT1 and recombinant DNMT1 were measured by EpiQuik DNMT Activity/Inhibition ELISA Easy Kit (P-3139, EpiGentek, Farmingdale, NY) according to the manufacturer's instructions. DNMT1 methylates DNA substrates by transferring a methyl group from AdoMet to cytosine, and the methylated DNA can be captured by anti-5-mC antibodies. The endogenous DNMT1 (by anti-DNMT1 Ab, 60B1220.1) and recombinant DNMT1 (by anti-Myc, ab18185) were enriched using immunoprecipitation from each cell or tissue lysates. DNMT1s were isolated and normalized by BCA analysis. The activity of 5 ng DNMT1 was analyzed by 450 nm ELISA with an optimal wavelength of 655 nm in a microplate spectrophotometer. The activity of the DNMT enzyme is proportional to the measured optical density intensity.

## Global DNA methylation LINE-1 assay

Global DNA methylation of LINE-1 elements was measured via kit (Active Motif, Cat #55017, Carlsbad, CA) according to the manufacturer's instructions. Hep3B and Myc-DNMT1-overexpressed mutants (DNMT1-WT or DNMT1-S878A) were treated with 5 mM glucose or 25 mM glucose, and 5-aza or GSK-3484862 (negative control). The activity of 100 ng was analyzed by 450 nm ELISA with an optimal wavelength of 655 nm.

## Agilent 4200 TapeStation

Global DNA methylation were measured by Agilent 4200 TapeStation system (Santa Clara, CA) with the Genomic DNA ScreenTape (5064-5365) and Genomic DNA Reagent (5067-5366) according to the manufacturer's instructions. Hep3B and Myc-DNMT1-overexpressed mutants (DNMT1-WT or DNMT1-S878A) were treated with 5 mM glucose or 25 mM glucose and 5-aza (negative control).

## Nanopore PromethION sequencing

Genomic DNA was isolated from DNMT1-WT or DNMT1-S878A cells treated with low (5 mM, CTRL) or high glucose/TMG (25 mM, *O*-GlcNAc) using the QIAGEN DNA Mini Kit (13323, QIAGEN) with Genomic-tip (10223, QIAGEN) according to the manufacturer's instructions. Sequencing libraries were prepared using the Ligation Sequencing Kit (SQK-LSK109, Oxford Nanopore Technologies) according to the manufacturer's instructions. Sequencing was performed on a PromethION (Oxford Nanopore Technologies). Data indexing was performed using Nanopolish (RRID:SCR_016157). Reads were aligned using minimap2 (RRID:SCR_018550) with the following parameters: -a -x map-ont (*Li and Birol, 2018*). Methylation state of CpGs was called using Nanopolish (RRID:SCR_016157) with the options call-methylation -t 8 (*Loman et al., 2015*). Only loci with >5× coverage were retained for analysis, comprising 90% of CpGs in the genome (*Figure 4—figure supplement 8*). Methylation percentage was averaged across CpG islands.

## Determination of gene-poor or gene-rich regions and FPKM

Hep3B RNA-seq data were obtained from *Chang et al., 2014*. Fastq files were aligned using HISAT2 version 2.1.0 (RRID:SCR_015530) to the hg19 genome. Duplications were removed using picard version 2.10.1 (RRID:SCR_006525). Aligned reads were sorted using samtools version 1.10 (RRID:SCR_002105). UCSC genome browser tracks were established using bedGraphToBigWig. Fragments Per Kilobase of transcript per Million mapped reads (FPKM) were calculated using StringTie version 1.3.4d (RRID:SCR_016323).

For all datasets, Bedgraph files were generated using bedtools version 2.29.0 (RRID:SCR_006646). BigWigs were generated using the UCSCtools bedGraphToBigWig. Heatmaps of global DNA methylation for DNMT1-WT and DNMT1-S878A cells under low or high glucose were generated using a custom script to profile the read coverage at each base and were visualized using pheatmap version 1.0.12 (RRID:SCR_016418). All other heatmaps and aggregate plots were generated using deeptools (RRID:SCR_016366).

## Measurement of ROS and PI staining

DNMT1-WT and DNMT1-S878A cells were seeded into 12-well plates with 5 mM or 25 mM glucose and TMG. The medium in each well was replaced with HBSS with 10 µM DCF-DA (2',7'-dichlorofluorescein diacetate; D6883, Abcam) or PI staining solution (P4864, Abcam). The fluorescence was filtered with fluorescein isothiocyanate (FITC) for ROS or Texas Red for PI staining (*Shin et al., 2020*). Averages of fluorescence were analyzed using Olympus Cellsens software.

## DNA damage analysis

DNA damage was analyzed with γ-H2A.X in DNMT1-WT and DNMT1-S878A cells and low (5 mM, CTRL) or high glucose/TMG (25 mM, *O*-GlcNAc) by immunofluorescence and using the EpiQuik 8-OHdG DNA Damage Quantification Direct Kit (P-6003, EpiGentek) according to the manufacturer's protocol. The activity of 200 ng DNA was analyzed with a 450 nm absorbance microplate reader.

## Apoptosis array analysis

Apoptosis-related proteins were analyzed using Proteome profiler human apoptosis array kit (ARY009, R&D Systems) according to the manufacturer's protocol. The spots were detected using a ChemiDoc MP Imaging system (Bio-Rad Laboratories; *Na et al., 2020*). The band intensity was densitometrically evaluated using Image Lab software (version 5.2, Bio-Rad Laboratories).

## KZFP binding in PMD-associated LTR12Cs

A list of PMDs in Hep3B cells were obtained from *Li et al., 2016*. A full list of LTR12Cs was generated from filtering RepeatMasker (*Smit et al., 2013*). The PMD-associated LTR12Cs were found using bedtools version 2.29.0 (RRID:SCR_006646). Putative KZFP regulators of LTR12Cs were determined using the consensus sequence of LTR12Cs and ChIP-exo from the Imbeault and Trono studies (*Imbeault et al., 2017*) on the UCSC Repeat Browser (*Fernandes et al., 2020*). PMD-associated LTR12Cs were then aligned again with peak files containing the ChIP-exo data to acquire a list of PMD-associated LTR12Cs with KZFP binding. Significant binding was defined as >5 sequences bound

as most LTR12Cs demonstrated very minimal KZFP binding (<1 sequence bound). These were verified with Repeat Browser (*Fernandes et al., 2020*).

## Quantification and statistical analysis

Statistical analyses were performed and graphed using GraphPad Prism 9 (v9.3.1). All statistical tests were performed by three independent biological replicate experiments assay, and the data are presented as means ± standard deviations. $*p<0.001$; $**p<0.0005$; $***p<0.0001$ by Student's $t$-test; ns, not significant; data are represented as mean ± SD.

## Acknowledgements

This article is dedicated to the memory of Dr. Arthur Riggs. This work was supported by the National Institutes of Health, grants R01DK112041 and R01CA220693 (DES), as well as a pilot award from the Arthur Riggs Diabetes & Metabolism Research Institute (DES). Research reported in this publication included work performed in the Pathology, Integrated Mass Spectrometry, and Integrative Genomics Shared Resources of the City of Hope, and was supported by the City of Hope CCSG Pilot award from the National Cancer Institute of the National Institutes of Health under award number P30CA033572.

## Additional information

### Funding

| Funder | Grant reference number | Author |
| --- | --- | --- |
| National Institutes of Health | R01DK112041 | Dustin E Schones |
| National Institutes of Health | R01CA220693 | Dustin E Schones |
| AR-DMRI City of Hope | Pilot | Dustin E Schones |

The funders had no role in study design, data collection and interpretation, or the decision to submit the work for publication.

### Author contributions

Heon Shin, Conceptualization, Data curation, Software, Formal analysis, Supervision, Funding acquisition, Validation, Investigation, Visualization, Methodology, Writing - original draft, Project administration, Writing - review and editing; Amy Leung, Conceptualization, Data curation, Software, Formal analysis, Validation, Investigation, Visualization, Methodology, Writing - original draft, Project administration, Writing - review and editing; Kevin R Costello, Conceptualization, Formal analysis, Investigation, Methodology, Writing - review and editing; Parijat Senapati, Hiroyuki Kato, Roger E Moore, Patrick Pirrotte, Formal analysis, Investigation; Michael Lee, Formal analysis, Investigation, Methodology, Writing - review and editing; Dimitri Lin, Formal analysis, Methodology, Writing - review and editing; Xiaofang Tang, Resources, Formal analysis; Zhen Bouman Chen, Resources; Dustin E Schones, Conceptualization, Resources, Formal analysis, Supervision, Funding acquisition, Investigation, Visualization, Methodology, Writing - original draft, Project administration, Writing - review and editing

### Author ORCIDs

Heon Shin http://orcid.org/0000-0001-5480-8492
Parijat Senapati http://orcid.org/0000-0002-7324-1230
Zhen Bouman Chen http://orcid.org/0000-0002-3291-1090
Dustin E Schones http://orcid.org/0000-0001-7692-8583

### Ethics

All animal experiments conducted have been approved by the Institutional Animal Care and Use Committees at City of Hope. All of the animals were handled according to approved institutional animal care and use committee (IACUC) protocols (#17010).

Decision letter and Author response
Decision letter https://doi.org/10.7554/eLife.85595.sa1
Author response https://doi.org/10.7554/eLife.85595.sa2

## Additional files

### Supplementary files
- Supplementary file 1. Prediction of *O*-GlcNAcylated sites within DNMT1 using OGTSite.
- Supplementary file 2. List of total identified proteins.
- Supplementary file 3. List of post-translational modification (PTM) sites of human DNMT1.
- Supplementary file 4. List of deposited data in this study.
- MDAR checklist

### Data availability
PromethION sequencing data have been deposited in the NCBI Gene Expression Omnibus (GEO) and Sequence Read Archive (SRA) under accession no. GSE201470. Mass spectrometry proteomics data have been deposited to the ProteomeXchange Consortium via the PRIDE partner repository with the data set identifier PXD043031.

The following datasets were generated:

| Author(s) | Year | Dataset title | Dataset URL | Database and Identifier |
| --- | --- | --- | --- | --- |
| Schones DE | 2023 | Inhibition of DNMT1 methyltransferase activity via glucose-regulated O-GlcNAcylation alters the epigenome | http://www.ncbi.nlm.nih.gov/geo/query/acc.cgi?acc=GSE201470 | NCBI Gene Expression Omnibus, GSE201470 |
| Schones DE | 2023 | Inhibition of DNMT1 methyltransferase activity via glucose-regulated O-GlcNAcylation alters the epigenome | https://www.ebi.ac.uk/pride/archive/projects/PXD043031 | PRIDE, PXD043031 |

The following previously published datasets were used:

| Author(s) | Year | Dataset title | Dataset URL | Database and Identifier |
| --- | --- | --- | --- | --- |
| Li X, Liu Y, Salz T, Hansen KD | 2016 | Whole-genome analysis of the methylome and hydroxymethylome in normal and malignant lung and liver | https://www.ncbi.nlm.nih.gov/geo/query/acc.cgi?acc=GSE70091 | NCBI Gene Expression Omnibus, GSE70091 |
| Chang C, Cui Y, Gu W, He Q, Wang T, Zhang G | 2014 | mRNA and RNC-mRNA deep sequencing of three hepatocellular carcinoma cell lines | https://www.ncbi.nlm.nih.gov/geo/query/acc.cgi?acc=GSE49994 | NCBI Gene Expression Omnibus, GSE49994 |

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
