## [Editor Report]

In this study Shin and colleagues investigate the role of the posttranslational modification of the DNA methyltransferase by covalent linkage of the N-Acetylglucosamine (O-GlcNAc). The authors present compelling evidence showing that a prolonged high fat/sucrose diet causes global protein O-GlcNAcylation in the liver and DNMT1 is among the proteins that increase their O-GlcNAc level. This result is significant because of the paucity of in vivo data addressing the interplay between metabolism and protein O-GlcNAcylation. The paper also shows that DNMT1's O-GlcNAcylation level correlated to extracellular glucose levels in other cell types.

---

## [Decision Letter]

**Decision letter after peer review:**

Thank you for submitting your article "Inhibition of DNMT1 methyltransferase activity via glucose-regulated O-GlcNAcylation alters the epigenome" for consideration by *eLife*. Your article has been reviewed by 3 peer reviewers, and the evaluation has been overseen by Carlos Isales as the Senior and Reviewing Editor. The following individual involved in the review of your submission has agreed to reveal their identity: Matthieu Boulard (Reviewer #1).

Essential revisions:

1) It would be very interesting to analyse the conservation of the DNMT1 O-GlcNAc sites and perhaps include the relevant protein alignment in Figure 2.

2) I fail to understand the reason to display the predicted sites of O-GlcNAcylation that were not confirmed experimentally.

3) It would be informative to mention the previously reported sites of phosphorylation of DNMT1.

4) Molecular weight should be indicated on western blots.

5) p-value/statistics are missing in Figure 4-supplement 3.

6) Figure 3E shows that treatment of cells with 25 mM glucose + TMG causes a substantial loss of DNA methylation at LINE1 elements (about a 20% decrease). Figure 3 sup 2 shows the result of the digestion of gDNA by the methyl-sensitive enzyme HpaII. While the demethylation in 5-aza treated cells is clear, the global methylation level cells cultured in presence of 25mM glucose + TMG is comparable to that of the control (C2), in contradiction with the Figure 3E that shows important demethylation in 25mM glucose + TMG condition. The discrepancy suggests to me that the global DNA methylation LINE-1 kit may not be very reliable.

7) Figure 4A actually shows a substantial loss of methylation in DNMT1-S878A mutant (CTRL) as compared to WT (CTRL), again suggesting that the global DNA methylation LINE-1 kit may not be very reliable.

8) The DNMT1 inhibitor used, namely 5-azacytidine has notorious cytotoxicity. Recently a new inhibitor developed by GlaxoSmithKline (e.g. GSK-3484862) was shown to be more effective with minimal non-specific toxicity and is therefore recommended (https://doi.org/10.1186/s13072-021-00429-0).

9) Figure 4—figure supplement 6: The evolutionary age should be made more explicit in the figure.

10) DNA methylation heat maps (Figure 5—figure supplement 1). The unit should be indicated on the color scheme (5mCpG/CpG).

11) As pointed out in the public review, the effect of TMG should be also tested in physiological glucose conditions.

12) The causal findings are based on a model of hyper-O-GlcNAcylation that increases O-GlcNAc on DNMT1 and also impacts many other proteins. It would be complementary to study the phenotype of the S878A mutation on endogenous DNMT1. Because S878 can also be phosphorylated, the result would inform on the regulatory function of S878 (via O-GlcNAc or phosphorylation).

13) An alternative approach to the induction of O-GlcNAcylation by TMG treatment is the genetic substitution S->C mutation that creates a thio-linked GlcNAc (S-GlcNAc), which is a non-hydrolyzable O-GlcNAc analog. https://www.nature.com/articles/s41594-019-0325-8

This trick could enable the study of the phenotype of hyperglycosylated DNMT1 more specifically (as TMG may have side effects and impact a large number of GlcNAc sites).

14) About "Here, we report that the activity of DNMT1 is regulated by extracellular levels of glucose through O-GlcNAcylation, resulting in loss of methylation within PMDs." (lines 54-56). The use of the word "resulting" is an overstatement because it is unknown whether a high glucose level causes loss of DNA methylation (the data shows that the combination of high glucose and OGA inhibitor causes loss of methylation but it remains to be tested if that's true with high glucose alone).

15) It would be interesting to speculate in the discussion on the possible mechanism of regulation of DNMT1 activity by S878-O-GlcNAc.

16) Quality and representation of the data

Many of the graphs are incorrectly labeled. It is incorrect to say "relative expression OglcNac/DNMT1" as in Figure 1A or 1B. The correct term is "relative abundance". It is definitely wrong to say "Relative expression (Cell death)" as in Figure 5C.

(17) Many of the graphs (eg 1A) have three dots for WT, all at value 1. I understand their value was set to 1. They also have 3 dots in the experimental condition. Typically, the variability is extremely small, making me think these are technical replicates with the same sample, instead of biological replicates. Also, which experimental point goes with which control?

(18) None of the IP experiments show inputs. I found this surprising at best, and misleading at worst.

(19) The assays for measuring DNMT activity are mostly ELISA assays. One has to look quite hard outside of the paper to understand how these assays work. For instance: does the DNMT1 Elisa assay start with hemimethylated DNA? This is critical for interpretation, but the information is not provided.

(20) The authors would do well to complement the ELISA assays with orthogonal methods such as LC-MS, enzymatic digestions, or others.

(21) Limitations of the experimental system

A large proportion of the conclusions are derived from the overexpression of DNMT1 (WT or mutant forms) in cell lines. Never are we shown or large or small the overexpression level is, even though it could be a major source of artifacts. It is clear that a better experimental solution would be a CRISPR knock-in mutation of S878A. Transient overexpression feels quite outdated.

(22) Overinterpretation

I feel that many conclusions are not solidly proven. For instance, we do not know for absolutely sure that OGlcNacylation of S878 decreases activity, as this position can also be phosphorylated. A more convincing experiment would be to purify the recombinant enzyme, OglcNac it in vitro, and test the activity in vitro.

(23) The most speculative part of the data is the last. Do the authors suggest that high glucose causes high OglcNac on DNMT1, causes low activity, causes gene misexpression, and causes ROS and apoptosis? How can we rule out that, instead, high glucose causes ROS, which causes apoptosis, which messes up DNA methylation artifactually?

---

## [Author Response]

Essential revisions:1) It would be very interesting to analyse the conservation of the DNMT1 O-GlcNAc sites and perhaps include the relevant protein alignment in Figure 2.

We thank the reviewers for suggesting this. In the revised manuscript we have compared the conservation of the DNMT1 *O*-GlcNAc sites between vertebrates. The modification site of human DNMT1 (S878) is conserved among vertebrates. We have included this in the revised manuscript (see page 6 and Figure 2B).

2) I fail to understand the reason to display the predicted sites of O-GlcNAcylation that were not confirmed experimentally.

In the revised manuscript we have updated this figure to show only the *O*-GlcNAcylated and phosphorylated sites of DNMT1 detected in this study. Several of the residues of DNMT1 that we found phosphorylated were recently added to the PRIDE database – those were also removed from the revised manuscript (see page 6 and Figure 2B).

3) It would be informative to mention the previously reported sites of phosphorylation of DNMT1.

There are 26 reported phosphorylated sites within DNMT1 (S35, S127, S133, T137, S141, S143, S152, S154, T166, S189, S192, S294, S312, S394, S398, Y399, S509, S549, S714, S732, S878, S953, S954, Y969, S1105, and S1122). The previously unreported phosphorylated sites that we have identified are T208, S209, and S1122. This information has been included in a new Supplemental Table (Table S3).

4) Molecular weight should be indicated on western blots.

We have added the molecular weight of western blots to all figures in the revised manuscript.

5) p-value/statistics are missing in Figure 4-supplement 3.

We apologize for this oversight. This has been corrected in the revised manuscript. This figure is now Figure 4—figure supplement 4.

6) Figure 3E shows that treatment of cells with 25 mM glucose + TMG causes a substantial loss of DNA methylation at LINE1 elements (about a 20% decrease). Figure 3 sup 2 shows the result of the digestion of gDNA by the methyl-sensitive enzyme HpaII. While the demethylation in 5-aza treated cells is clear, the global methylation level cells cultured in presence of 25mM glucose + TMG is comparable to that of the control (C2), in contradiction with the Figure 3E that shows important demethylation in 25mM glucose + TMG condition. The discrepancy suggests to me that the global DNA methylation LINE-1 kit may not be very reliable.

We apologize for not making this clear in the initial submission. We have now detailed the observed difference in genomic DNA methylation levels between the LINE-1 kit and HpaII digestion/gel separation of genomic DNA – to state they both have similar trends in loss of DNA in the revised manuscript but the observed differences may be due to the differences in digestions between MspI and HpaII. The analysis from direct sequencing utilizing the PromethION has also shown loss of DNA methylation at LINE elements suggesting our overall conclusion that loss of DNA methylation generally occurs is valid (Figure 4 —figure supplement 4). We have moved Figure 3E to Figure 3—figure supplement 2A.

7) Figure 4A actually shows a substantial loss of methylation in DNMT1-S878A mutant (CTRL) as compared to WT (CTRL), again suggesting that the global DNA methylation LINE-1 kit may not be very reliable.

As indicated above, the revised manuscript includes a more detailed explanation.

8) The DNMT1 inhibitor used, namely 5-azacytidine has notorious cytotoxicity. Recently a new inhibitor developed by GlaxoSmithKline (e.g. GSK-3484862) was shown to be more effective with minimal non-specific toxicity and is therefore recommended (https://doi.org/10.1186/s13072-021-00429-0).

We thank the reviewer for this suggestion. We have utilized this inhibitor in the revised manuscript alongside the 5-aza analysis (Figure3—figure supplement 2A ; Figure 5 —figure supplement 4).

9) Figure 4—figure supplement 6: The evolutionary age should be made more explicit in the figure.

We added evolutionary distance with the age of each group. The evolutionary distance between each group is from Perelman et al., PLoS Genet 2011.

Perelman, P., Johnson, W.E., Roos, C., Seuanez, H.N., Horvath, J.E., Moreira, M.A., Kessing, B., Pontius, J., Roelke, M., Rumpler, Y., et al. (2011). A molecular phylogeny of living primates. PLoS Genet 7, e1001342.

10) DNA methylation heat maps (Figure 5—figure supplement 1). The unit should be indicated on the color scheme (5mCpG/CpG).

We apologize for this oversight. We have corrected this in the revised manuscript (Figure 4D and Figure 5—figure supplement 1).

11) As pointed out in the public review, the effect of TMG should be also tested in physiological glucose conditions.

We thank the reviewers for suggesting these analyses to strengthen the manuscript. The physiological glucose levels are between 5 to 7 mM, and 25mM is in hyperglycemic range, which corresponds to severe diabetes. The new Figure 1A shows TMG treatment with physiological glucose conditions. We have included new WB data of 5mM glucose, 5mM glucose + TMG, 25mM glucose, and 25mM glucose + TMG (Figure 1A).

12) The causal findings are based on a model of hyper-O-GlcNAcylation that increases O-GlcNAc on DNMT1 and also impacts many other proteins. It would be complementary to study the phenotype of the S878A mutation on endogenous DNMT1. Because S878 can also be phosphorylated, the result would inform on the regulatory function of S878 (via O-GlcNAc or phosphorylation).

We have toned down the language throughout the revised manuscript. We demonstrated *O*-GlcNAcylation of DNMT1 using IP/WB and mass spec analysis. An alanine mutation was performed to block the posttranslational modification of serine (Hardwick et al., 1986). In addition, it was demonstrated that the DNA methyltransferase activity of DNMT1 due to phospho-S878 was not affected through the use of S878D (May et al., 1998). The DNMT1 activity assay in Figure3D utilized only the myc-tagged recombinant DNMT1. Through this it was verified that the activity of DNMT1 decreased only in the high glucose + Thiamet-G treated condition. We have furthermore revised Figure 3—figure supplement 3A, B using DNMT1-S878D to verify that phosphorylation of DNMT1 within S878 does not affect DNA methyltransferase activity.

Hardwick, J.M., Shaw, K.E., Wills, J.W., and Hunter, E. (1986). Amino-terminal deletion mutants of the Rous sarcoma virus glycoprotein do not block signal peptide cleavage but can block intracellular transport. J Cell Biol 103, 829-838.

May, G.H., Allen, K.E., Clark, W., Funk, M., and Gillespie, D.A. (1998). Analysis of the interaction between c-Jun and c-Jun N-terminal kinase in vivo. J Biol Chem 273, 33429-33435.

13) An alternative approach to the induction of O-GlcNAcylation by TMG treatment is the genetic substitution S->C mutation that creates a thio-linked GlcNAc (S-GlcNAc), which is a non-hydrolyzable O-GlcNAc analog. https://www.nature.com/articles/s41594-019-0325-8. This trick could enable the study of the phenotype of hyperglycosylated DNMT1 more specifically (as TMG may have side effects and impact a large number of GlcNAc sites).

We thank the reviewer for this interesting suggestion. As indicated, side effects may exist because TMG impacts proteins globally. The current project is focused on the change in the function of DNMT1 according to the presence or absence of *O*-GlcNAc in S878 by the change in glucose level. In other words, after blocking other external factors in normal and hyperglycemic conditions, only the post-translational modification of DNMT1 is changed and then its function is determined, which is considered to be of sufficient value in terms of protein function change. Also, as shown in Figure 4A, the substitution of one amino acid in DNMT1 could cause a slight decrease in the DNA methyltransferase function of DNMT1. Considering this, substitution with cysteine could also affect DNMT1. This is indeed an interesting avenue of study and one we intend to follow up on in future work.

14) About "Here, we report that the activity of DNMT1 is regulated by extracellular levels of glucose through O-GlcNAcylation, resulting in loss of methylation within PMDs." (lines 54-56). The use of the word "resulting" is an overstatement because it is unknown whether a high glucose level causes loss of DNA methylation (the data shows that the combination of high glucose and OGA inhibitor causes loss of methylation but it remains to be tested if that's true with high glucose alone).

We have toned down the language throughout the revised manuscript.

15) It would be interesting to speculate in the discussion on the possible mechanism of regulation of DNMT1 activity by S878-O-GlcNAc.

We have included discussion on possible mechanisms of regulation of DNMT1 activity in the revised manuscript.

16) Quality and representation of the dataMany of the graphs are incorrectly labeled. It is incorrect to say "relative expression OglcNac/DNMT1" as in Figure 1A or 1B. The correct term is "relative abundance". It is definitely wrong to say "Relative expression (Cell death)" as in Figure 5C.

We apologize for these oversights. They have been resolved in the revised manuscript.

(17) Many of the graphs (eg 1A) have three dots for WT, all at value 1. I understand their value was set to 1. They also have 3 dots in the experimental condition. Typically, the variability is extremely small, making me think these are technical replicates with the same sample, instead of biological replicates. Also, which experimental point goes with which control?

We thank the reviewer for the suggestion. Most of our experiments were performed in 3 biological replicates. For ease of viewing, each control was set to 1 (for example, 5 mM low glucose treated Hep3B cells in Figure 1A). However, the display has been changed to show more information in the revised manuscript. We furthermore provided more details to the text to indicate these are biological replicates (page 24).

(18) None of the IP experiments show inputs. I found this surprising at best, and misleading at worst.

We apologize for this oversight. We have included these controls in the revised manuscript.

(19) The assays for measuring DNMT activity are mostly ELISA assays. One has to look quite hard outside of the paper to understand how these assays work. For instance: does the DNMT1 Elisa assay start with hemimethylated DNA? This is critical for interpretation, but the information is not provided.

We thank the reviewer for the suggestion. We included more details in the Methods in the revised manuscript (page 20).

(20) The authors would do well to complement the ELISA assays with orthogonal methods such as LC-MS, enzymatic digestions, or others.

We thank the reviewer for the suggestion. As the reviewer pointed out, for Figure 3 and Figure 5B, ELISA methods were performed. In the case of measuring the DNA methyltransferase activity of Figure 3, we examined several different kits for measuring methyltransferase activity, but all of them were ELISA based. Measurement of oxidative DNA damage using 8-OHdG in Figure 5B was further bolstered with fluorescence data using γH2A.X.

(21) Limitations of the experimental systemA large proportion of the conclusions are derived from the overexpression of DNMT1 (WT or mutant forms) in cell lines. Never are we shown or large or small the overexpression level is, even though it could be a major source of artifacts. It is clear that a better experimental solution would be a CRISPR knock-in mutation of S878A. Transient overexpression feels quite outdated.

The mutation from serine to phenylalanine at amino acid S878 of DNMT1 was recently reported in β-thalassemia (Gong et al., 2021). This work demonstrates that the mutation abolishes the phosphorylation of DNMT1, resulting in low stability and loss of catalytic activity. This indicates that damage to the S878 site of DNMT1 inhibits DNA methyltransferase activity. As shown in Figure 4A, there is a slight loss of methylation in the S878A mutant despite the low glucose concentration, indicating a potential compromise of function for the overexpressed protein. The CRISPR knock-in experiments are a great suggestion but are not without their own potential sources of error. In particular, using CRISPR/Cas9 in the field of *O*-GlcNAc research can be challenging. For example, it has been reported that sugar modifications caused by microscopic protein structural changes can change the aggregation behavior of the protein α-synuclein (Moon et al., 2022; Galesic et al., 2021). This is something we intend to pursue in follow up work.

Gong, Y., Zhang, X., Zhang, Q., Zhang, Y., Ye, Y., Yu, W., Shao, C., Yan, T., Huang, J., Zhong, J., et al. (2021). A natural DNMT1 mutation elevates the fetal hemoglobin level via epigenetic derepression of the γ-globin gene in β-thalassemia. Blood *137*, 1652-1657.

Moon, S.P., Javed, A., Hard, E.R., and Pratt, M.R. (2022). Methods for Studying Site-Specific O-GlcNAc Modifications: Successes, Limitations, and Important Future Goals. JACS Au 2, 74-83.

Galesic, A., Rakshit, A., Cutolo, G., Pacheco, R.P., Balana, A.T., Moon, S.P., and Pratt, M.R. (2021). Comparison of N-Acetyl-Glucosamine to Other Monosaccharides Reveals Structural Differences for the Inhibition of α-Synuclein Aggregation. ACS Chem Biol 16, 14-19.

(22) OverinterpretationI feel that many conclusions are not solidly proven. For instance, we do not know for absolutely sure that OGlcNacylation of S878 decreases activity, as this position can also be phosphorylated. A more convincing experiment would be to purify the recombinant enzyme, OglcNac it in vitro, and test the activity in vitro.

We have toned down the language throughout the revised manuscript. Our finding with the DNMT1-S878D mutant (Figure 3D), demonstrates that the DNA methyltransferase activity of DNMT1 due to phospho-S878 mimetic does that recapture the effects of DNMT1-S878A. (Hardwick et al., 1986; May et al., 1998).

Hardwick, J.M., Shaw, K.E., Wills, J.W., and Hunter, E. (1986). Amino-terminal deletion mutants of the Rous sarcoma virus glycoprotein do not block signal peptide cleavage but can block intracellular transport. J Cell Biol 103, 829-838.

May, G.H., Allen, K.E., Clark, W., Funk, M., and Gillespie, D.A. (1998). Analysis of the interaction between c-Jun and c-Jun N-terminal kinase in vivo. J Biol Chem 273, 33429-33435.

(23) The most speculative part of the data is the last. Do the authors suggest that high glucose causes high OglcNac on DNMT1, causes low activity, causes gene misexpression, and causes ROS and apoptosis? How can we rule out that, instead, high glucose causes ROS, which causes apoptosis, which messes up DNA methylation artifactually?

We thank the reviewer for the suggestion. According to our results, DNMT1-S878A mutant suppressed the observed DNA damage and cell death due to high glucose/TMG treatment suggesting that elevated DNMT1 *O*-GlcNAcylation and subsequent reduction in its activity influences DNA damage and cell death. To supplement this observation, we have now included additional experiments with GSK-3484862 showing that loss of DNMT1 activity can increase DNA damage. We have further toned down language with regards to these results.